# Information-theoretic generalization bounds for black-box learning algorithms

**Hrayr Harutyunyan**[1]**, Maxim Raginsky**[2]**, Greg Ver Steeg**[1]**, Aram Galstyan**[1]
[1] USC Information Sciences Institute, [2] University of Illinois Urbana-Champaign

## Abstract

We derive information-theoretic generalization bounds for supervised learning algorithms based on the information contained in predictions rather than in the output of the training algorithm. These bounds improve over the existing information-theoretic bounds, are applicable to a wider range of algorithms, and solve two key challenges: (a) they give meaningful results for deterministic algorithms and (b) they are significantly easier to estimate. We show experimentally that the proposed bounds closely follow the generalization gap in practical scenarios for deep learning.

## 1 Introduction

Large neural networks trained with variants of stochastic gradient descent have excellent generalization capabilities, even in regimes where the number of parameters is much larger than the number of training examples. Zhang et al. [41] showed that classical generalization bounds based on various notions of complexity of hypothesis set fail to explain this phenomenon, as the same neural network can generalize well for one choice of training data and memorize completely for another one. This observation has spurred a tenacious search for algorithm-dependent and data-dependent generalization bounds that give meaningful results in practical settings for deep learning [17].

One line of attack bounds generalization error based on the information about training dataset stored in the weights [39, 4, 22, 6, 33, 14, 23, 27]. The main idea is that when the training and testing performance of a neural network are different, the network weights necessarily capture some information about the training dataset. However, the opposite might not be true: A neural network can store significant portions of training set in its weights and still generalize well [32, 40, 21]. Furthermore, because of their information-theoretic nature, these generalization bounds become infinite or produce trivial bounds for deterministic algorithms. When such bounds are not infinite, they are notoriously hard to estimate, due to the challenges arising in estimation of Shannon mutual information between two high-dimensional variables (e.g., weights of a ResNet and a training dataset).

This work addresses the aforementioned challenges. We first improve some of the existing information-theoretic generalization bounds, providing a unified view and derivation of them (Sec. 2). We then derive novel generalization bounds that measure information with predictions, rather than with the output of the training algorithm (Sec. 3). These bounds are applicable to a wide range of methods, including neural networks, Bayesian algorithms, ensembling algorithms, and non-parametric approaches. In the case of neural networks, the proposed bounds improve over the existing weight-based bounds, partly because they avoid a counter-productive property of weight-based bounds that information stored in *unused* weights affects generalization bounds, even though it has no effect on generalization. The proposed bounds produce meaningful results for deterministic algorithms and are significantly easier to estimate. For example, in case of classification, computing our most efficient bound involves estimating mutual information between a pair of predictions and a binary variable.

35th Conference on Neural Information Processing Systems (NeurIPS 2021).

We apply the proposed bounds to ensembling algorithms, binary classification algorithms with finite VC dimension hypothesis classes, and to stable learning algorithms (Sec. 4). We compute our most efficient bound on realistic classification problems involving neural networks, and show that the bound closely follows the generalization error, even in situations when a neural network with 3M parameters is trained deterministically on 4000 examples, achieving 1% generalization error.

## 2    Weight-based generalization bounds

We start by describing the necessary notation and definitions, after which we present some of the existing weigh-based information-theoretic generalization bounds, slightly improve some of them, and prove relations between them. The purpose of this section is to introduce the relevant existing bounds and prepare grounds for the functional conditional mutual information bounds introduced in Sec. 3, which we consider our main contribution. All proofs are presented in Appendix A.

**Preliminaries.**    We use capital letters for random variables, corresponding lowercase letters for their values, and calligraphic letters for their domains. If $X$ is a random variable, $\bar{X}$ denotes an independent copy of $X$. For example, if $(X, Y)$ is a pair of random variables with joint distribution $P_{X,Y}$, then the joint distribution of $(\bar{X}, \bar{Y})$ will be $P_{\bar{X},\bar{Y}} = P_{\bar{X}} \otimes P_{\bar{Y}} = P_X \otimes P_Y$. A random variable $X$ is called $\sigma$-subgaussian if $\mathbb{E} \exp(t(X - \mathbb{E} X)) \leq \exp(\sigma^2 t^2/2)$, $\forall t \in \mathbb{R}$. For example, a random variable that takes values in $[a, b]$ almost surely, is $(b - a)/2$-subgaussian. Given probability measures $P$ and $Q$ defined on the same measurable space, such that $P$ is absolutely continuous with respect to $Q$, the Kullback–Leibler divergence from $P$ to $Q$ is defined as $\text{KL}(P \parallel Q) = \int \log \frac{dP}{dQ} dP$, where $\frac{dP}{dQ}$ is the Radon-Nikodym derivative of $P$ with respect to $Q$. If $X$ and $Y$ are random variables defined on the same probability space, then $\text{KL}(X \parallel Y)$ denotes $\text{KL}(P_X \parallel P_Y)$. The Shannon mutual information between random variables $X$ and $Y$ is $I(X; Y) = \text{KL}(P_{X,Y} \parallel P_X \otimes P_Y)$. In this paper, all information-theoretic quantities are measured in nats, instead of bits. Throughout the paper $[n]$ denotes the set $\{1, 2, \ldots, n\}$. Finally, if $A = (a_1, \ldots, a_n)$ is a collection, then $A_{-i} \triangleq (a_1, \ldots, a_{i-1}, a_{i+1}, \ldots, a_n)$.

Theorems proved in the subsequent sections will be relying on the following lemma.

**Lemma 1.**    *Let $(\Phi, \Psi)$ be a pair of random variables with joint distribution $P_{\Psi,\Phi}$. If $g(\phi, \psi)$ is a measurable function such that $\mathbb{E}_{\Phi,\Psi}[g(\Phi, \Psi)]$ exists and $g(\bar{\Phi}, \bar{\Psi})$ is $\sigma$-subgaussian, then*

$$\left| \mathbb{E}_{\Phi,\Psi}[g(\Phi, \Psi)] - \mathbb{E}_{\Phi,\bar{\Psi}}[g(\Phi, \bar{\Psi})] \right| \leq \sqrt{2\sigma^2 I(\Phi; \Psi)}. \tag{1}$$

*Furthermore, if $g(\phi, \bar{\Psi})$ is $\sigma$-subgaussian for each $\phi$ and the expectation below exists, then*

$$\mathbb{E}_{\Phi,\Psi}\left[ \left( g(\Phi, \Psi) - \mathbb{E}_{\bar{\Psi}} g(\Phi, \bar{\Psi}) \right)^2 \right] \leq 4\sigma^2 (I(\Phi; \Psi) + \log 3), \tag{2}$$

*and*

$$P\left( \left| g(\Phi, \Psi) - \mathbb{E}_{\bar{\Psi}} g(\Phi, \bar{\Psi}) \right| \geq \epsilon \right) \leq \frac{4\sigma^2 (I(\Phi; \Psi) + \log 3)}{\epsilon^2}, \quad \forall \epsilon > 0. \tag{3}$$

The first part of this lemma is equivalent to Lemma 1 of Xu and Raginsky [39], which in turn has its roots in Russo and Zou [29]. The second part generalizes Lemma 2 of Hafez-Kolahi et al. [13] by also providing bounds on the expected squared difference.

### 2.1    Generalization bounds with input-output mutual information

Let $S = (Z_1, Z_2, \ldots, Z_n) \sim \mathcal{D}^n$ be a dataset of $n$ i.i.d. examples, $R \in \mathcal{R}$ be a source of randomness (a random variable independent of $S$) and $A : \mathcal{Z}^n \times \mathcal{R} \to \mathcal{W}$ be a training algorithm. Let $W = A(S, R)$ be the output of the training algorithm applied on the dataset $S$ with randomness $R$. Given a loss function $\ell : \mathcal{W} \times \mathcal{Z} \to \mathbb{R}$, the empirical risk is $\mathcal{L}_{\text{emp}}(A, S, R) = \frac{1}{n} \sum_{i=1}^{n} \ell(W, Z_i)$ and the population risk is $\mathcal{L}(A, S, R) = \mathbb{E}_{Z' \sim D} \ell(W, Z')$, where $Z'$ is a test example independent from $S$ and $R$. The generalization gap, also call generalization error, is $\mathcal{L}(A, S, R) - \mathcal{L}_{\text{emp}}(A, S, R)$. In this setting, Xu and Raginsky [39] establish the following information-theoretic bound on the absolute value of the expected generalization gap.

**Theorem 2.1** (Thm. 1 of Xu and Raginsky [39]). *If $\ell(w, Z')$, where $Z' \sim \mathcal{D}$, is $\sigma$-subgaussian for all $w \in \mathcal{W}$, then*

$$|\mathbb{E}_{S,R} [\mathcal{L}(A, S, R) - \mathcal{L}_{\text{emp}}(A, S, R)]| \leq \sqrt{\frac{2\sigma^2 I(W; S)}{n}}. \tag{4}$$

We generalize this result by showing that instead of measuring information with the entire dataset, one can measure information with a subset of size $m$ chosen uniformly at random. For brevity, hereafter we call subsets chosen uniformly at random just "random subsets".

**Theorem 2.2.** *Let $U$ be a random subset of $[n]$ with size $m$, independent of $S$ and $R$. If $\ell(w, Z')$, where $Z' \sim \mathcal{D}$, is $\sigma$-subgaussian for all $w \in \mathcal{W}$, then*

$$|\mathbb{E}_{S,R} [\mathcal{L}(A, S, R) - \mathcal{L}_{\text{emp}}(A, S, R)]| \leq \mathbb{E}_{u \sim U} \sqrt{\frac{2\sigma^2}{m} I(W; S_u)}, \tag{5}$$

*and*

$$\mathbb{E}_{S,R} (\mathcal{L}(A, S, R) - \mathcal{L}_{\text{emp}}(A, S, R))^2 \leq \frac{4\sigma^2}{n} (I(W; S) + \log 3). \tag{6}$$

With a simple application of Markov's inequality one can get tail bounds from the second part of the theorem. Furthermore, by taking square root of both sides of (6) and using Jensen's inequality on the left side, one can also construct an upper bound for the expected absolute value of generalization gap, $\mathbb{E}_{S,R} |\mathcal{L}(A, S, R) - \mathcal{L}_{\text{emp}}(A, S, R)|$. These observations apply also to the other generalization gap bounds presented later in this work.

Note the bound on the squared generalization gap is written only for the case of $m = n$. It is possible to derive squared generalization gap bounds of form $\frac{4\sigma^2}{m}(\mathbb{E}_{u \sim U} I(W; S_u) + \log 3)$. Unfortunately, for small $m$ the $\log 3$ constant starts to dominate, resulting in vacuous bounds.

Picking a small $m$ decreases the mutual information term in (5), however, it also decreases the denominator. When setting $m = n$, we get the bound of Xu and Raginsky [39] (Thm. 2.1). When $m = 1$, the bound of (5) becomes $\frac{1}{n} \sum_{i=1}^{n} \sqrt{2\sigma^2 I(W; Z_i)}$, matching the result of Bu et al. [6] (Proposition 1). A similar bound, but for a different notion of information, was derived by Alabdulmohsin [2]. Bu et al. [6] prove that the bound with $m = 1$ is tighter than the bound with $m = n$. We generalize this result by proving that the bound of (5) is non-decreasing in $m$.

**Proposition 1.** *Let $m \in [n-1]$, $U$ be a random subset of $[n]$ of size $m$, $U'$ be a random subset of size $m + 1$, and $\phi : \mathbb{R} \to \mathbb{R}$ be any non-decreasing concave function. Then*

$$\mathbb{E}_U \phi \left( \frac{1}{m} I(W; S_u) \right) \leq \mathbb{E}_{U'} \phi \left( \frac{1}{m+1} I(W; S_{u'}) \right). \tag{7}$$

When $\phi(x) = \sqrt{x}$, this result proves that the optimal value for $m$ in (5) is 1. Furthermore, when we use Jensen's inequality to move expectation over $U$ inside the square root in (5), then the resulting bound becomes $\sqrt{\frac{2\sigma^2}{m} \mathbb{E}_{u \sim U} I(W; S_u)}$ and matches the result of Negrea et al. [22] (Thm. 2.3). These bounds are also non-decreasing with respect to $m$ (using Proposition 1 with $\phi(x) = x$).

Thm. 2.1 can be used to derive generalization bounds that depend on the information between $W$ and a single example $Z_i$ conditioned on the remaining examples $Z_{-i} = (Z_1, \dots, Z_{i-1}, Z_{i+1}, \dots, Z_n)$.

**Theorem 2.3.** *If $\ell(w, Z')$, where $Z' \sim \mathcal{D}$, is $\sigma$-subgaussian for all $w \in \mathcal{W}$, then*

$$|\mathbb{E}_{S,R} [\mathcal{L}(A, S, R) - \mathcal{L}_{\text{emp}}(A, S, R)]| \leq \frac{1}{n} \sum_{i=1}^{n} \sqrt{2\sigma^2 I(W; Z_i \mid Z_{-i})}, \tag{8}$$

*and*

$$\mathbb{E}_{S,R} (\mathcal{L}(A, S, R) - \mathcal{L}_{\text{emp}}(A, S, R))^2 \leq \frac{4\sigma^2}{n} \left( \sum_{i=1}^{n} I(W; Z_i \mid Z_{-i}) + \log 3 \right). \tag{9}$$

This theorem is a simple corollary of Thm. 2.2, using the facts that $I(W; Z_i) \leq I(W; Z_i \mid Z_{-i})$ and that $I(W; S)$ is upper bounded by $\sum_{i=1}^n I(W; Z_i \mid Z_{-i})$, which is also known as erasure information [35]. The first part of it improves the result of Raginsky et al. [26] (Thm. 2), as the averaging over $i$ is outside of the square root. While these bounds are worse that the corresponding bounds of Thm. 2.2, it is sometimes easier to manipulate them analytically.

The bounds described above measure information with the output $W$ of the training algorithm. In the case of prediction tasks with parametric methods, the parameters $W$ might contain information about the training dataset, but not use it to make predictions. Partly for this reason, the main goal of this paper is to derive generalization bounds that measure information with the prediction function, rather than with the weights. In general, there is no straightforward way of encoding the prediction function into a random variable. However, when the domain $\mathcal{Z}$ is finite, we can encode the prediction function as the collection of predictions on all examples of $\mathcal{Z}$. This naturally leads us to the next setting (albeit with a different motivation), first considered by Steinke and Zakynthinou [33], where one first fixes a set of $2n$ examples, and then randomly selects $n$ of them to form the training set. We use this setting to provide prediction-based generalization bounds in Sec. 3. Before describing these bounds we present the setting of Steinke and Zakynthinou [33] in detail and generalize some of the existing weight-based bounds in that setting.

## 2.2 Generalization bounds with conditional mutual information

Let $\tilde{Z} \in \mathcal{Z}^{n \times 2}$ be a collection of $2n$ i.i.d samples from $\mathcal{D}$, grouped in $n$ pairs. The random variable $S \sim \text{Uniform}(\{0, 1\}^n)$ specifies which example to select from each pair to form the training set $\tilde{Z}_S = (\tilde{Z}_{i, S_i})_{i=1}^n$. Let $R$ be a random variable, independent of $\tilde{Z}$ and $S$, that captures the stochasticity of training. In this setting Steinke and Zakynthinou [33] defined condition mutual information (CMI) of algorithm $A$ with respect to the data distribution $\mathcal{D}$ as

$$\text{CMI}_{\mathcal{D}}(A) = I(A(\tilde{Z}_S, R); S \mid \tilde{Z}) = \mathbb{E}_{\tilde{z} \sim \tilde{Z}} I(A(\tilde{z}_S, R); S), \tag{10}$$

and proved the following upper bound on expected generalization gap.

**Theorem 2.4** (Thm. 2, Steinke and Zakynthinou [33]). *If the loss function* $\ell(w, z) \in [0, 1], \forall w \in \mathcal{W}, z \in \mathcal{Z}$, *then the expected generalization gap can be bounded as follows:*

$$\left| \mathbb{E}_{\tilde{Z}, S, R} \left[ \mathcal{L}(A, \tilde{Z}_S, R) - \mathcal{L}_{\text{emp}}(A, \tilde{Z}_S, R) \right] \right| \leq \sqrt{\frac{2}{n} \text{CMI}_{\mathcal{D}}(A)}. \tag{11}$$

Haghifam et al. [14] improved this bound in two aspects. First, they provided bounds where expectation over $\tilde{Z}$ is outside of the square root. Second, they considered measuring information with subsets of $S$, as we did in the previous section.

**Theorem 2.5** (Thm. 3.1 of Haghifam et al. [14]). *Let* $m \in [n]$ *and* $U \subseteq [n]$ *be a random subset of size* $m$, *independent from* $R, \tilde{Z}$, *and* $S$. *If the loss function* $\ell(w, z) \in [0, 1], \forall w \in \mathcal{W}, z \in \mathcal{Z}$, *then*

$$\left| \mathbb{E}_{\tilde{Z}, S, R} \left[ \mathcal{L}(A, \tilde{Z}_S, R) - \mathcal{L}_{\text{emp}}(A, \tilde{Z}_S, R) \right] \right| \leq \mathbb{E}_{z \sim \tilde{Z}} \sqrt{\frac{2}{m} \mathbb{E}_{u \sim U} I(A(\tilde{z}_S, R); S_u)}. \tag{12}$$

Furthermore, for $m = 1$ they tighten the bound by showing that one can move the expectation over $U$ outside of the squared root (Haghifam et al. [14], Thm 3.4). We generalize these results by showing that for all $m$ expectation over $U$ can be done outside of the square root. Furthermore, our proof closely follows the proof of Thm. 2.2.

**Theorem 2.6.** *Let* $m \in [n]$ *and* $U \subseteq [n]$ *be a random subset of size* $m$, *independent from* $R, \tilde{Z}$, *and* $S$. *If* $\ell(w, z) \in [0, 1], \forall w \in \mathcal{W}, z \in \mathcal{Z}$, *then*

$$\left| \mathbb{E}_{\tilde{Z}, S, R} \left[ \mathcal{L}(A, \tilde{Z}_S, R) - \mathcal{L}_{\text{emp}}(A, \tilde{Z}_S, R) \right] \right| \leq \mathbb{E}_{\tilde{z} \sim \tilde{Z}, u \sim U} \sqrt{\frac{2}{m} I(A(\tilde{z}_S, R); S_u)}, \tag{13}$$

*and*

$$\mathbb{E}_{\tilde{Z}, S, R} \left( \mathcal{L}(A, \tilde{Z}_S, R) - \mathcal{L}_{\text{emp}}(A, \tilde{Z}_S, R) \right)^2 \leq \frac{8}{n} \left( \mathbb{E}_{\tilde{z} \sim \tilde{Z}} I(A(\tilde{z}_S, R); S) + 2 \right). \tag{14}$$

The bound of (13) improves over the bound of Thm. 2.5 and matches the special result for $m = 1$. Rodríguez-Gálvez et al. [28] proved even tighter expected generalization gap bound by replacing $I(A(\tilde{Z}_S, R); S_u \mid \tilde{Z} = \tilde{z})$ with $I(A(\tilde{Z}_S, R); S_u \mid \tilde{Z}_u = \tilde{z}_u)$. Haghifam et al. [14] showed that if one takes the expectations over $\tilde{Z}$ inside the square root in (12), then the resulting looser upper bounds become non-decreasing over $m$. Using this result they showed that their special case bound for $m = 1$ is the tightest. We generalize their results by showing that even without taking the expectations inside the squared root, the bounds of Thm. 2.5 are non-decreasing over $m$. We also show that the same holds for our tighter bounds of (13).

**Proposition 2.** *Let $m \in [n-1]$, $U$ be a random subset of $[n]$ of size $m$, $U'$ be a random subset of size $m + 1$, $\tilde{z}$ be any fixed value of $\tilde{Z}$, and $\phi : \mathbb{R} \to \mathbb{R}$ be any non-decreasing concave function. Then*

$$\mathbb{E}_{u \sim U} \, \phi \left( \frac{1}{m} I(A(\tilde{z}_S, R); S_u) \right) \leq \mathbb{E}_{u' \sim U'} \, \phi \left( \frac{1}{m+1} I(A(\tilde{z}_S, R); S_{u'}) \right). \tag{15}$$

By setting $\phi(x) = x$, taking square root of both sides of (15), and then taking expectation over $\tilde{z}$, we prove that bounds of (12) are non-decreasing over $m$. By setting $\phi(x) = \sqrt{x}$ and then taking expectation over $\tilde{z}$, we prove that bounds of (13) are non-decreasing with $m$.

Similarly to the Thm. 2.3 of the previous section, Thm. A.1 presented in Appendix A establishes generalization bounds with information-theoretic stability quantities.

# 3 Functional conditional mutual information

The bounds in Sec. 2 leverage information in the output of the algorithm, $W$. In this section we focus on supervised learning problems: $\mathcal{Z} = \mathcal{X} \times \mathcal{Y}$. To encompass many types of approaches, we do not assume that the training algorithm has an output $W$, which is then used to make predictions. Instead, we assume that the learning method implements a function $f : \mathcal{Z}^n \times \mathcal{X} \times \mathcal{R} \to \mathcal{K}$ that takes a training set $z$, a test input $x'$, an auxiliary argument $r$ capturing the stochasticity of training and predictions, and outputs a prediction $f(z, x', r)$ on the test example. Note that the prediction domain $\mathcal{K}$ can be different from $\mathcal{Y}$. This setting includes non-parametric methods (for which $W$ is the training dataset itself), parametric methods, Bayesian algorithms, and more. For example, in parametric methods, where a hypothesis set $\mathcal{H} = \{h_w : \mathcal{X} \to \mathcal{K} \mid w \in \mathcal{W}\}$ is defined, $f(z, x, r) = h_{A(z,r)}(x)$.

In this supervised setting, the loss function $\ell : \mathcal{K} \times \mathcal{Y} \to \mathbb{R}$ measures the discrepancy between a prediction and a label. As in the previous subsection, we assume that a collection of $2n$ i.i.d examples $\tilde{Z} \sim \mathcal{D}^{n \times 2}$ is given, grouped in $n$ pairs, and the random variable $S \sim \text{Uniform}(\{0, 1\}^n)$ specifies which example to select from each pair to form the training set $\tilde{Z}_S = (\tilde{Z}_{i,S_i})_{i=1}^n$. Let $R$ be an auxiliary random variable, independent of $\tilde{Z}$ and $S$, that provides stochasticity for predictions (e.g., in neural networks $R$ can be used to make the training stochastic). The empirical risk of learning method $f$ trained on dataset $\tilde{Z}_S$ with randomness $R$ is defined as $\mathcal{L}_{\text{emp}}(f, \tilde{Z}_S, R) = \frac{1}{n} \sum_{i=1}^n \ell(f(\tilde{Z}_S, X_i, R), Y_i)$. The population risk is defined as $\mathcal{L}(f, \tilde{Z}_S, R) = \mathbb{E}_{Z' \sim \mathcal{D}} \ell(f(\tilde{Z}_S, X', R), Y')$. Before moving forward we adopt two conventions. First, if $z$ is a collection of examples, then $x$ and $y$ denote the collection of its inputs and labels respectively. Second, if $x$ is a collection of inputs, then $f(z, x, r)$ denotes the collection of predictions on $x$ after training on $z$ with randomness $r$.

We define functional conditional mutual information ($f$-CMI).

**Definition 3.1.** *Let $\mathcal{D}$, $f$, $R$, $\tilde{Z}$, $S$ be defined as above and let $u \subseteq [n]$ be a subset of size $m$. Then pointwise functional conditional mutual information $f$-CMI$(f, \tilde{z}, u)$ is defined as*

$$f\text{-CMI}(f, \tilde{z}, u) = I(f(\tilde{z}_S, \tilde{x}_u, R); S_u), \tag{16}$$

*while functional conditional mutual information $f$-CMI$_\mathcal{D}(f, u)$ is defined as*

$$f\text{-CMI}_\mathcal{D}(f, u) = \mathbb{E}_{\tilde{z} \sim \tilde{Z}} \, f\text{-CMI}(f, \tilde{z}, u). \tag{17}$$

When $u = [n]$ we will simply use the notations $f$-CMI$(f, \tilde{z})$ and $f$-CMI$_\mathcal{D}(f)$, instead of $f$-CMI$(f, \tilde{z}, [n])$ and $f$-CMI$_\mathcal{D}(f, [n])$, respectively.

**Theorem 3.1.** *Let $U$ be a random subset of size $m$, independent of $\tilde{Z}$, $S$, and randomness of training algorithm $f$. If $\ell(\hat{y}, y) \in [0, 1], \forall \hat{y} \in \mathcal{K}, z \in \mathcal{Z}$, then*

$$\left| \mathbb{E}_{\tilde{Z}, R, S} \left[ \mathcal{L}(f, \tilde{Z}_S, R) - \mathcal{L}_{\text{emp}}(f, \tilde{Z}_S, R) \right] \right| \leq \mathbb{E}_{\tilde{z} \sim \tilde{Z}, u \sim U} \sqrt{\frac{2}{m} f\text{-CMI}(f, \tilde{z}, u)}, \tag{18}$$

*and*

$$\mathbb{E}_{\tilde{Z},R,S} \left( \mathcal{L}(f, \tilde{Z}_S, R) - \mathcal{L}_{\text{emp}}(f, \tilde{Z}_S, R) \right)^2 \leq \frac{8}{n} \left( \mathbb{E}_{\tilde{z} \sim \tilde{Z}} \, f\text{-CMI}(f, \tilde{z}) + 2 \right). \qquad (19)$$

For parametric methods, the bound of (18) improves over the bound of (13), as the Markov chain $S_u$ — $A(\tilde{z}_S, R)$ — $f(\tilde{z}_S, \tilde{x}_u, R)$ allows to use the data processing inequality $I(f(\tilde{z}_S, \tilde{x}_u, R); S_u) \leq I(A(\tilde{z}_S, R); S_u)$. For deterministic algorithms $I(A(\tilde{z}_S); S_u)$ is often equal to $H(S_u) = m \log 2$, as most likely each choice of $S$ produces a different $W = A(\tilde{z}_S)$. In such cases the bound with $I(W; S_u)$ is vacuous. In contrast, the proposed bounds with $f$-CMI (especially when $m = 1$) do not have this problem. Even when the algorithm is stochastic, information between $W$ and $S_u$ can be much larger than information between predictions and $S_u$, as having access to weights makes it easier to determine $S_u$ (e.g., by using gradients). A similar phenomenon has been observed in the context of membership attacks, where having access to weights of a neural network allows constructing more successful membership attacks compared to having access to predictions only [21, 12].

**Corollary 1.** *When $m = n$, the bound of (18) becomes*

$$\left| \mathbb{E}_{\tilde{Z},R,S} \left[ \mathcal{L}(f, \tilde{Z}_S, R) - \mathcal{L}_{\text{emp}}(f, \tilde{Z}_S, R) \right] \right| \leq \mathbb{E}_{\tilde{z} \sim \tilde{Z}} \sqrt{\frac{2}{n} f\text{-CMI}(f, \tilde{z})} \leq \sqrt{\frac{2}{n} f\text{-CMI}_{\mathcal{D}}(f)}. \quad (20)$$

For parametric models, this improves over the CMI bound (Thm. 2.4), as by data processing inequality, $f\text{-CMI}_{\mathcal{D}}(f) = I(f(\tilde{Z}_S, \tilde{X}, R); S \mid \tilde{Z}) \leq I(A(\tilde{Z}_S, R); S \mid \tilde{Z}) = \text{CMI}_{\mathcal{D}}(A)$.

**Remark 1.** Note that the collection of training and testing predictions $f(\tilde{Z}_S, \tilde{X}, R)$ cannot be replaced with only testing predictions $f(\tilde{Z}_S, \tilde{X}_{\text{neg}(S)}, R)$. As an example, consider an algorithm that memorizes the training examples and outputs a constant prediction on any other example. This algorithm will have non-zero generalization gap, but $f(\tilde{Z}_S, \tilde{X}_{\text{neg}(S)}, R)$ will be constant and will have zero information with $S$ conditioned on any random variable. Moreover, if we replace $f(\tilde{Z}_S, \tilde{X}, R)$ with only training predictions $f(\tilde{Z}_S, \tilde{X}_S, R)$, the resulting bound can become too loose, as one can deduce $S$ by comparing training set predictions with the labels $\tilde{Y}$.

**Corollary 2.** *When $m = 1$, the bound of (18) becomes*

$$\left| \mathbb{E}_{\tilde{Z},R,S} \left[ \mathcal{L}(f, \tilde{Z}_S, R) - \mathcal{L}_{\text{emp}}(f, \tilde{Z}_S, R) \right] \right| \leq \frac{1}{n} \sum_{i=1}^{n} \mathbb{E}_{\tilde{z} \sim \tilde{Z}} \sqrt{2I(f(\tilde{z}_S, \tilde{x}_i, R); S_i)}. \qquad (21)$$

A great advantage of this bound compared to all other bounds described so far is that the mutual information term is computed between a relatively low-dimensional random variable $f(\tilde{z}_S, \tilde{x}_i, R)$ and a binary random variable $S_i$. For example, in the case of binary classification with $\mathcal{K} = \{0, 1\}$, $f(\tilde{z}_S, \tilde{x}_i, R)$ will be a pair of 2 binary variables. This allows us to estimate the bound efficiently and accurately (please refer to Appendix B for more details). Note that estimating other information-theoretic bounds is significantly harder. The bounds of Xu and Raginsky [39], Negrea et al. [22], and Bu et al. [6] are hard to estimate as they involve estimation of mutual information between a high-dimensional non-discrete variable $W$ and at least one example $Z_i$. Furthermore, this mutual information can be infinite in case of deterministic algorithms or when $H(Z_i)$ is infinite. The bounds of Haghifam et al. [14] and Steinke and Zakynthinou [33] are also hard to estimate as they involve estimation of mutual information between $W$ and at least one train-test split variable $S_i$.

As in the case of bounds presented in the previous section (Thm. 2.2 and Thm. 2.6), we prove that the bound of Thm. 3.1 is non-decreasing in $m$. This stays true even when we increase the upper bounds by moving the expectation over $U$ or the expectation over $\tilde{Z}$ or both under the square root. The following proposition allows us to prove all these statements.

**Proposition 3.** *Let $m \in [n-1]$, $U$ be a random subset of $[n]$ of size $m$, $U'$ be a random subset of size $m + 1$, $\tilde{z}$ be any fixed value of $\tilde{Z}$, and $\phi : \mathbb{R} \rightarrow \mathbb{R}$ be any non-decreasing concave function. Then*

$$\mathbb{E}_{u \sim U} \, \phi \left( \frac{1}{m} I(f(\tilde{z}_S, \tilde{x}_u, R); S_u) \right) \leq \mathbb{E}_{u' \sim U'} \, \phi \left( \frac{1}{m+1} I(f(\tilde{z}_S, \tilde{x}_{u'}, R); S_{u'}) \right). \qquad (22)$$

By setting $\phi(x) = \sqrt{x}$ and then taking expectation over $\tilde{z}$ and $u$, we prove that bounds of Thm. 3.1 are non-decreasing over $m$. By setting $\phi(x) = x$, taking expectation over $\tilde{z}$, and then taking square root of both sides of (22), we prove that bounds are non-decreasing in $m$ when both expectations are under the square root. Proposition 3 proves that $m = 1$ is the optimal choice in Thm. 3.1. Notably, the bound that is the easiest to compute is also the tightest!

Analogously to Thm. A.1, we provide the following stability-based bounds.

**Theorem 3.2.** *If* $\ell(\widehat{y}, y) \in [0, 1], \forall \widehat{y} \in \mathcal{K}, z \in \mathcal{Z}$, *then*

$$\left| \mathbb{E}_{\tilde{Z}, R, S} \left[ \mathcal{L}(f, \tilde{Z}_S, R) - \mathcal{L}_{\text{emp}}(f, \tilde{Z}_S, R) \right] \right| \leq \mathbb{E}_{\tilde{z} \sim \tilde{Z}} \left[ \frac{1}{n} \sum_{i=1}^{n} \sqrt{2I(f(\tilde{z}_S, \tilde{x}_i, R); S_i \mid S_{-i})} \right], \quad (23)$$

*and*

$$\mathbb{E}_{\tilde{Z}, R, S} \left( \mathcal{L}(f, \tilde{Z}_S, R) - \mathcal{L}_{\text{emp}}(f, \tilde{Z}_S, R) \right)^2 \leq \frac{8}{n} \left( \mathbb{E}_{\tilde{z} \sim \tilde{Z}} \left[ \sum_{i=1}^{n} I(f(\tilde{z}_S, \tilde{x}, R); S_i \mid S_{-i}) \right] + 2 \right).$$

Note that unlike (23), in the second part of Thm. 3.2 we measure information with predictions on all $2n$ pairs and $S_i$ conditioned on $S_{-i}$. It is an open question whether $f(\tilde{z}_S, \tilde{x}, R)$ can be replaced with $f(\tilde{z}_S, \tilde{x}_i, R)$ – predictions only on the $i$-th pair.

# 4 Applications

In this section we describe 3 applications of the $f$-CMI-based generalization bounds.

## 4.1 Ensembling algorithms

Ensembling algorithms combine predictions of multiple learning algorithms to obtain better performance. Let us consider $k$ learning algorithms, $f_1, f_2, \ldots, f_k$, each with its own independent randomness $R_i, \ i \in [k]$. Some ensembling algorithms can be viewed as a possibly stochastic function $g : \mathcal{K}^k \to \mathcal{K}$ that takes predictions of the $k$ algorithms and combines them into a single prediction. Relating the generalization gap of the resulting ensembling algorithm to that of individual $f_i$s can be challenging for complicated choices of $g$. However, it is easy to bound the generalization gap of $g(f_1, \ldots, f_k)$ in terms of $f$-CMIs of individual predictors. Let $\tilde{z}$ be a fixed value of $\tilde{Z}$ and $x$ be an arbitrary collection of inputs. Denoting $F_i = f_i(\tilde{z}_S, x, R_i), \ i \in [k]$, we have that

$$I(g(F_1, \ldots, F_k); S) \leq I(F_1, \ldots, F_k; S) \qquad \text{(data processing inequality)}$$
$$= I(F_1; S) + I(F_2, \ldots, F_k; S) - I(F_1; F_2, \ldots, F_k) + I(F_1; F_2, \ldots, F_k \mid S) \quad \text{(chain rule)}$$
$$\leq I(F_1; S) + I(F_2, \ldots, F_k; S) \qquad \text{(as MI is nonnegative and } F_1 \perp\!\!\!\perp F_2, \ldots, F_k \mid S)$$
$$\leq \ldots \leq I(F_1; S) + \cdots + I(F_k; S). \qquad \text{(repeating the arguments above to separate all } F_i)$$

Unfortunately, the same derivation above does not work if we replace $S$ with $S_u$, where $u$ is a proper subset of $[n]$, as $I(F_1; F_2, \ldots, F_k \mid S_u)$ will not be zero in general.

## 4.2 Binary classification with finite VC dimension

Let us consider the case of binary classification: $\mathcal{Y} = \{0, 1\}$, where the learning method $f : \mathcal{Z}^n \times \mathcal{X} \times \mathcal{R} \to \{0, 1\}$ is implemented using a learning algorithm $A : \mathcal{Z}^n \times \mathcal{R} \to \mathcal{W}$ that selects a classifier from a hypothesis set $\mathcal{H} = \{h_w : \mathcal{X} \to \mathcal{Y}\}$. If $\mathcal{H}$ has finite VC dimension $d$ [34], then for any algorithm $f$, the quantity $f$-CMI$(f, \tilde{z})$ can be bounded the following way.

**Theorem 4.1.** *Let* $\mathcal{Z}, \mathcal{H}, f$ *be defined as above, and let* $d < \infty$ *be the VC dimension of* $\mathcal{H}$. *Then for any algorithm* $f$ *and* $\tilde{z} \in \mathcal{Z}^{n \times 2}$,

$$f\text{-CMI}(f, \tilde{z}) \leq \max \{(d + 1) \log 2, \ d \log (2en/d)\}. \quad (24)$$

Considering the 0-1 loss function and using this result in Corollary 1, we get an expect generalization gap bound that is $O\left(\sqrt{\frac{d}{n} \log\left(\frac{n}{d}\right)}\right)$, matching the classical uniform convergence bound [34]. The $\sqrt{\log n}$ factor can be removed in some cases [13].

Both Xu and Raginsky [39] and Steinke and Zakynthinou [33] prove similar information-theoretic bounds in the case of finite VC dimension classes, but their results holds for specific algorithms only. Even in the simple case of threshold functions: $\mathcal{X} = [0,1]$ and $\mathcal{H} = \left\{h_w : x \mapsto \mathbf{1}_{\{x>w\}} \mid w \in [0,1]\right\}$, all weight-based bounds described in Sec. 2 are vacuous if one uses a training algorithm that encodes the training set in insignificant bits of $W$, while still getting zero error on the training set and hence achieving low test error.

## 4.3 Stable deterministic or stochastic algorithms

Theorems 2.3, A.1 and 3.2 provide generalization bounds involving information-theoretic stability measures, such as $I(W; Z_i \mid Z_{-i})$, $I(A(\tilde{z}_S, R); S \mid S_{-i})$ and $I(f(\tilde{z}_S, \tilde{x}, R); S_i \mid S_{-i})$. In this section we build upon the predication-based stability bounds of Thm. 3.2. First, we show that for any collection of examples $x$, the mutual information $I(f(\tilde{z}_S, x); S_i \mid S_{-i})$ can be bounded as follows.

**Proposition 4.** *Let $S^{i \leftarrow c}$ denote $S$ with $S_i$ set to c. Then for any $\tilde{z} \in \mathcal{Z}^{n \times 2}$ and $\tilde{x} \in \mathcal{X}^k$, the mutual information $I(f(\tilde{z}_S, x, R); S_i \mid S_{-i})$ is upper bounded by*

$$\frac{1}{4}KL\left(f(\tilde{z}_{S^{i \leftarrow 1}}, x, R)|S_{-i} \parallel f(\tilde{z}_{S^{i \leftarrow 0}}, x, R)|S_{-i}\right) + \frac{1}{4}KL\left(f(\tilde{z}_{S^{i \leftarrow 0}}, x, R)|S_{-i} \parallel f(\tilde{z}_{S^{i \leftarrow 1}}, x, R)|S_{-i}\right).$$

To compute the right-hand side of Proposition 4 one needs to know how much on-average the distribution of predictions on $x$ changes after replacing the $i$-th example in the training dataset. The problem arises when we consider deterministic algorithms. In such cases, the right-hand side is infinite, while the left-hand side $I(f(\tilde{z}_S, x, R); S_i \mid S_{-i})$ is always finite and could be small. Therefore, for deterministic algorithms, directly applying the result of Proposition 4 will not give meaningful generalization bounds. Nevertheless, we show that we can add an optimal amount of noise to predictions, upper bound the generalization gap of the resulting noisy algorithm, and relate that to the generalization gap of the original deterministic algorithm.

Let us consider a deterministic algorithm $f : \mathcal{Z}^n \times \mathcal{X} \to \mathbb{R}^d$. We define the following notions of functional stability.

**Definition 4.1** (Functional stability). *Let $S = (Z_1, \dots, Z_n) \sim \mathcal{D}^n$ be a collection of $n$ i.i.d. samples, and $Z'$ and $Z_{test}$ be two additional independent samples from $\mathcal{D}$. Let $S^{(i)} \triangleq (Z_1, \dots, Z_{i-i}, Z', Z_{i+1}, \dots, Z_n)$ be the collection constructed from $S$ by replacing the $i$-th example with $Z'$. A deterministic algorithm $f : \mathcal{Z}^n \times \mathcal{X} \to \mathbb{R}^d$ is*

*a) $\beta$ self-stable if $\forall i \in [n]$, $\mathbb{E}_{S,Z'} \left\| f(S, Z_i) - f(S^{(i)}, Z_i) \right\|^2 \leq \beta^2$,* (25)

*b) $\beta_1$ test-stable if $\forall i \in [n]$, $\mathbb{E}_{S,Z',Z_{test}} \left\| f(S, Z_{test}) - f(S^{(i)}, Z_{test}) \right\|^2 \leq \beta_1^2$,* (26)

*c) $\beta_2$ train-stable if $\forall i, j \in [n], i \neq j$, $\mathbb{E}_{S,Z'} \left\| f(S, Z_j) - f(S^{(i)}, Z_j) \right\|^2 \leq \beta_2^2$.* (27)

**Theorem 4.2.** *Let $\mathcal{Y} = \mathbb{R}^d$, $f : \mathcal{Z}^n \times \mathcal{X} \to \mathbb{R}^d$ be a deterministic algorithm that is $\beta$ self-stable, and $\ell(\hat{y}, y) \in [0,1]$ be a loss function that is $\gamma$-Lipschitz in the first coordinate. Then*

$$\left| \mathbb{E}_{\tilde{Z}, R, S} \left[ \mathcal{L}(f, \tilde{Z}_S, R) - \mathcal{L}_{\text{emp}}(f, \tilde{Z}_S, R) \right] \right| \leq 2^{\frac{3}{2}} d^{\frac{1}{4}} \sqrt{\gamma \beta}. \tag{28}$$

*Furthermore, if $f$ is also $\beta_1$ train-stable and $\beta_2$ test-stable, then*

$$\mathbb{E}_{\tilde{Z}, R, S} \left( \mathcal{L}(f, \tilde{Z}_S, R) - \mathcal{L}_{\text{emp}}(f, \tilde{Z}_S, R) \right)^2 \leq \frac{32}{n} + 12^{\frac{3}{2}} \sqrt{d} \gamma \sqrt{2\beta^2 + n\beta_1^2 + n\beta_2^2}. \tag{29}$$

It is expected that $\beta_2$ is smaller than $\beta$ and $\beta_1$. For example, in the case of neural networks interpolating the training data or in the case of empirical risk minimization in the realizable setting, $\beta_2$ will be zero. It is also expected that $\beta$ is larger than $\beta_1$. However, the relation of $\beta^2$ and $n\beta_1^2$ is not trivial.

The notion of pointwise hypothesis stability $\beta_2'$ defined by Bousquet and Elisseeff [5] (definition 4) is comparable to our notion of self-stability $\beta$. The first part of Theorem 11 in [5] describes a generalization bound where the difference between empirical and population losses is of order

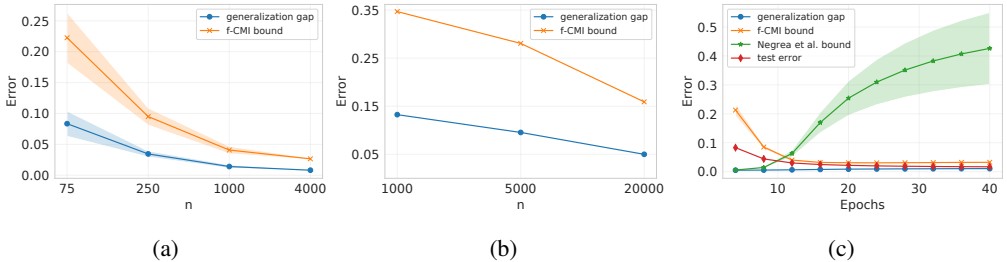

(a)                          (b)                          (c)

Figure 1: Comparison of expected generalization gap and $f$-CMI bound in 3 settings: (a) MNIST 4 vs 9 classification with a CNN trained using a deterministic algorithm, (b) pretrained ResNet-50 finetuned on CIFAR-10, and (c) MNIST 4 vs 9 with a CNN trained using SGLD.

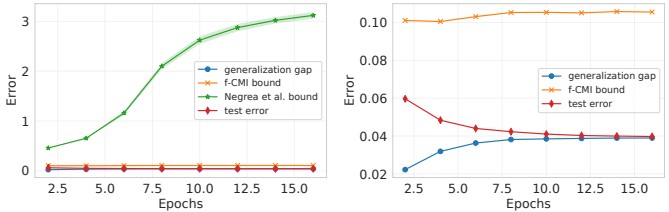

Figure 2: Comparison of expected generalization gap, Negrea et al. [22] SGLD bound and $f$-CMI bound in case of a pretrained ResNet-50 fine-tuned with SGLD on a subset of CIFAR-10 of size $n = 20000$. The figure on the right is the zoomed-in version of the figure on the left.

$1/\sqrt{n} + \sqrt{\beta_2'}$, which is comparable with our result of Thm. 4.2 ($\Theta(\sqrt{\beta})$). The proof there also contains a bound on the expected squared difference of empirical and population losses. That bound is of order $1/n + \beta_2'$. In contrast, our result of (29) contains two extra terms related to test-stability and train-stability (the terms $n\beta_1^2$ and $n\beta_2^2$). If $\beta$ dominates $n\beta_1^2 + n\beta_2^2$, then the bound of (29) will match the result of Bousquet and Elisseeff [5].

## 5  Experiments

As mentioned earlier, the expected generalization gap bound of Corollary 2 is significantly easier to compute compared to existing information-theoretic bounds, and does not give trivial results for deterministic algorithms. To understand how well the bound does in challenging situations, we consider cases when the algorithm generalizes well despite the high complexity of the hypothesis class and relatively small number of training examples. Due to space constraints we omit some experimental details and present them in Appendix B. The code can be found at github.com/hrayrhar/f-CMI.

First, we consider the MNIST 4 vs 9 digit classification task [20] using a 4-layer convolutional neural network (CNN) that has approximately 200K parameters. We train the network using for 200 epochs using the ADAM algorithm [18] with 0.001 learning rate, $\beta_1 = 0.9$, and mini-batches of 128 examples. Importantly, we fix the random seed that controls the initialization of weights and the shuffling of training data, making the training algorithm deterministic. Fig. 1a plots the expected generalization gap and the $f$-CMI bound of (21). We see that the bound is not vacuous and is not too far from the expected generalization gap even when considering only 75 training examples. As shown in the Fig. 3a of Appendix B, if we increase the width of all layers 4 times, making the number of parameters approximately 3M, the results remain largely unchanged.

Next, we move away from binary classification and consider the CIFAR-10 classication task [19]. To construct a well-generalizing algorithm, we use the ResNet-50 [16] network pretrained on the ImageNet [7], and fine-tune it for 40 epochs using SGD with mini-batches of size 64, 0.01 learning rate, 0.9 momentum, and standard data augmentations. The results presented in Fig. 1b indicate that the $f$-CMI bound is always approximately 3 times larger than the expected generalization gap. In particular, when $n = 20000$, the expected generalization gap is 5%, while the bound predicts 16%.

Note that the weight-based information-theoretic bounds discussed in Sec. 2 would give either infinite or trivial bounds for the deterministic algorithm described above. Even when we make the training algorithm stochastic by randomizing the seed, the quantities like $I(W; S)$ still remain infinite, while both the generalization gap and the $f$-CMI bound do not change significantly (see Fig. 3b of Appendix B). For this reason, we change the training algorithm to Stochastic Gradient Langevin Dynamics (SGLD) [10, 38] and compare the $f$-CMI-based bound against the specialized bound of Negrea et al. [22] (see eq. (6) of [22]). This bound (referred as SGLD bound here) is derived from a weight-based information-theoretic generalization bound, and depends on the the hyper-parameters of SGLD and on the variance of per-example gradients along the training trajectory. The SGLD algorithm is trained for 40 epochs, with learning rate and inverse temperature schedules described in Appendix B. Fig. 1c plots the expected generalization gap, the expected test error, the $f$-CMI bound and the SGLD bound. We see that the test accuracy plateaus after 16 epochs. At this time and afterwards, the $f$-CMI bound closely follows the generalization gap, while the SGLD bound increases to very high values. However, we see that the SGLD bound does better up to epoch 12.

The difference between the $f$-CMI bound and the SGLD bound becomes more striking when we change the dataset to be a subset of CIFAR-10 consisting of 20000 examples, and fine-tune a pretrained ResNet-50 with SGLD. As shown in Fig. 2, even after a single epoch the SGLD bound is approximately 0.45, while the generalization gap is around 0.02. For comparison, the $f$-CMI is approximately 0.1 after one epoch of training.

Interestingly, Fig. 1c shows that the f-CMI bound is large in the early epochs, despite of the extremely small generalization gap. In fact, a similar trend, albeit with lesser extent, is visible in the MNIST 4 vs 9 experiment, where a CNN is trained with a deterministic algorithm (see Fig. 3c of Appendix B). This indicates a possible area of improvement for the $f$-CMI bound.

# 6 Related Work

This work is closely related to a rich literature of information-theoretic generalization bounds, some of which were discussed earlier [39, 4, 25, 22, 6, 33, 14, 13, 1, 23, 27, 8]. Most of these work derive generalization bounds that depend on a mutual information quantity measured between the output of the training algorithm and some quantity related to training data. Different from this major idea, Xu and Raginsky [39] and Russo and Zou [29] discussed the idea of bounding generalization gap with the information between the input and the vector of loss functions computed on training examples. This idea was later extended to the setting of conditional mutual information by Steinke and Zakynthinou [33]. This works are similar to ours in the sense that they move away from measuring information with weights, but they did not develop this line of reasoning enough to arrive to efficient bounds similar to Corollary 2. Additionally, we believe that measuring information with the prediction function allows better interpretation and is easier to work with analytically.

Another related line of research are the stability-based bounds [5, 2, 26, 3, 9, 37, 27]. In Sec. 2 and Sec. 3 we improve existing generalization bounds that use information stability. In Sec. 4.3 we describe a technique of applying information stability bounds to deterministic algorithms. The main idea is to add noise to predictions, but only for analysis purposes. A similar idea, but in the context of measuring information content of an individual example, was suggested by Harutyunyan et al. [15]. In fact, our notion of test-stability defined in Sec. 4.3 comes very close to their definition of functional sample information. A similar idea was recently used by Neu et al. [23] in analyzing generalization performance of SGD. More broadly this work is related to PAC-Bayes bounds and to classical generalization bounds. Please refer to the the survey by Jiang* et al. [17] for more information on these bounds.

Finally, our work has connections with attribute and membership inference attacks [32, 40, 21, 12]. Some of these works show that having a white-box access to models allows constructing better membership inference attacks, compared to having a black-box access. This is analogous to our observation that prediction-based bounds are better than weight-based bounds. Shokri et al. [32] and Yeom et al. [40] demonstrate that even in the case of *black-box access to a well-generalizing* model, sometimes it is still possible to construct successful membership attacks. This is in line with our observation that the $f$-CMI bound can be significantly large, despite of small generalization gap (see epoch 4 of Fig. 1c). This suggests a possible direction of improving the $f$-CMI-based bounds.

## Acknowledgments and Disclosure of Funding

This work is based on research sponsored by Air Force Research Laboratory (AFRL) under agreement number FA8750-19-1-1000. The U.S. Government is authorized to reproduce and distribute reprints for Government purposes notwithstanding any copyright notation therein. The views and conclusions contained herein are those of the authors and should not be interpreted as necessarily representing the official policies or endorsements, either expressed or implied, of Air Force Laboratory, DARPA or the U.S. Government. HH was partially supported by a USC Annenberg Fellowship.

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
