# A    Additional statements and proofs

This appendix includes the missing proofs of the results presented in the main text, additional results, and some helpful lemmas.

**Fact 1** (Thm. 5.2.1 of Gray [11]). *Let $P$ and $Q$ be two probability measures defined on the same measurable space $(\Omega, \mathcal{F})$, such that $P$ is absolutely continuous with respect to $Q$. Then the Donsker-Varadhan dual characterization of Kullback-Leibler divergence states that*

$$KL\left(P \parallel Q\right) = \sup_{X} \left\{ \mathbb{E}_P\left[X\right] - \log \mathbb{E}_Q\left[e^X\right] \right\}, \tag{30}$$

*$X$ is any random variable defined on $(\Omega, \mathcal{F})$ such that both $\mathbb{E}_P\left[X\right]$ and $\mathbb{E}_Q\left[e^X\right]$ exist.*

**Lemma 2.** *If $X$ is a $\sigma$-subgaussian random variable with zero mean, then*

$$\mathbb{E}\,e^{\lambda X^2} \leq 1 + 8\lambda\sigma^2, \quad \forall \lambda \in \left[0, \frac{1}{4\sigma^2}\right). \tag{31}$$

*Proof.* As $X$ is $\sigma$-subgaussian and $\mathbb{E}\,X = 0$, the $k$-th moment of $X$ can be bounded the following way [35]:

$$\mathbb{E}\,|X|^k \leq (2\sigma^2)^{k/2} k\Gamma(k/2), \quad \forall k \in \mathbb{N}, \tag{32}$$

where $\Gamma(\cdot)$ is the Gamma function. Continuing,

$$\mathbb{E}\,e^{\lambda X^2} = \mathbb{E}\left[\sum_{k=0}^{\infty} \frac{(\lambda X^2)^k}{k!}\right] \tag{33}$$

$$= 1 + \sum_{k=1}^{\infty} \mathbb{E}\left(\frac{(\sqrt{\lambda}|X|)^{2k}}{k!}\right) \qquad \text{(by Fubini's theorem)} \tag{34}$$

$$\leq 1 + \sum_{k=1}^{\infty} \left(\frac{(2\lambda\sigma^2)^k \cdot 2k \cdot \Gamma(k)}{k!}\right) \qquad \text{(by (32))} \tag{35}$$

$$= 1 + 2\sum_{k=1}^{\infty} (2\lambda\sigma^2)^k. \tag{36}$$

When $\lambda \leq 1/(4\sigma^2)$, the infinite sum of (36) converges to a value that is at most twice of the first element of the sum. Therefore

$$\mathbb{E}\,e^{\lambda X^2} \leq 1 + 8\lambda\sigma^2, \quad \forall \lambda \in \left[0, \frac{1}{4\sigma^2}\right). \tag{37}$$

$\square$

**Lemma 3.** *Let $X$ and $Y$ be independent random variables. If $g$ is a measurable function such that $g(x, Y)$ is $\sigma$-subgaussian and $\mathbb{E}\,g(x, Y) = 0$ for all $x \in \mathcal{X}$, then $g(X, Y)$ is also $\sigma$-subgaussian.*

*Proof.* As $\mathbb{E}\,g(X, Y) = 0$, we have that

$$\mathbb{E}_{X,Y} \exp\left\{t\left(g(X,Y) - \mathbb{E}_{X,Y}g(X,Y)\right)\right\} = \mathbb{E}_{X,Y} \exp\left\{tg(X,Y)\right\} \tag{38}$$

$$= \mathbb{E}_{x\sim X}\left[\mathbb{E}_Y \exp\left\{tg(x,Y)\right\}\right] \qquad \text{(by independence of $X$ and $Y$)} \tag{39}$$

$$\leq \mathbb{E}_{x\sim X}\,e^{t^2\sigma^2} \qquad \text{(by subgaussianity of $g(x,Y)$)} \tag{40}$$

$$= e^{t^2\sigma^2}. \tag{41}$$

$\square$

## A.1 Proof of Lemma 1

We use the Donsker-Varadhan inequality (see Fact 1) for $I(\Phi; \Psi)$ and $\lambda g(\phi, \psi)$, where $\lambda \in \mathbb{R}$ is any constant:

$$I(\Phi; \Psi) = \mathrm{KL}\left(P_{\Phi, \Psi} \parallel P_\Phi \otimes P_\Psi\right) \qquad \text{(by definition)} \tag{42}$$

$$\geq \mathbb{E}_{\Phi, \Psi}\left[\lambda g(\Phi, \Psi)\right] - \log \mathbb{E}_{\bar\Phi, \bar\Psi}\left[e^{\lambda g(\bar\Phi, \bar\Psi)}\right] \qquad \text{(by Fact 1)} \tag{43}$$

$$= \mathbb{E}_{\Phi, \Psi}\left[\lambda g(\Phi, \Psi)\right] - \log \mathbb{E}_{\Phi, \bar\Psi}\left[e^{\lambda g(\Phi, \bar\Psi)}\right] \qquad \text{(as } P_{\bar\Phi, \bar\Psi} = P_{\Phi, \bar\Psi}). \tag{44}$$

The subgaussianity of $g(\Phi, \bar\Psi)$ implies that

$$\log \mathbb{E}_{\Phi, \bar\Psi}\left[e^{\lambda\left(g(\Phi, \bar\Psi) - \mathbb{E}[g(\Phi, \bar\Psi)]\right)}\right] \leq \frac{\lambda^2 \sigma^2}{2}, \quad \forall \lambda \in \mathbb{R}. \tag{45}$$

Plugging this into (44), we get that

$$I(\Phi; \Psi) \geq \lambda\left(\mathbb{E}_{\Phi, \Psi}\left[g(\Phi, \Psi)\right] - \mathbb{E}_{\Phi, \bar\Psi}\left[g(\Phi, \bar\Psi)\right]\right) - \frac{\lambda^2 \sigma^2}{2}. \tag{46}$$

Picking $\lambda$ to maximize the right-hand side, we get that

$$I(\Phi; \Psi) \geq \frac{1}{2\sigma^2}\left(\mathbb{E}_{\Phi, \Psi}\left[g(\Phi, \Psi)\right] - \mathbb{E}_{\Phi, \bar\Psi}\left[g(\Phi, \bar\Psi)\right]\right)^2, \tag{47}$$

which proves the first part of the lemma.

To prove the second part of the lemma, we are going to use Donsker-Varadhan inequality again, but for a different function. Let $\lambda \in \left[0, \frac{1}{4\sigma^2}\right)$ and define

$$\tilde{g}(\phi, \psi) \triangleq \lambda\left(g(\phi, \psi) - \mathbb{E}_{\bar\Psi}\, g(\phi, \bar\Psi)\right)^2. \tag{48}$$

By assumption $\mathbb{E}\,\tilde{g}(\Phi, \Psi)$ exists. Note that for each fixed $\phi$, the random variable $g(\phi, \bar\Psi) - \mathbb{E}_{\bar\Psi}\, g(\phi, \bar\Psi)$ has zero mean and is $\sigma$-subgaussian, by the additional assumptions of the second part of the lemma. As a result, $\mathbb{E}\exp\left(\tilde{g}(\Phi, \bar\Psi)\right) = \mathbb{E}_{\phi \sim \Phi}\,\mathbb{E}_{\bar\Psi}\exp\left(\tilde{g}(\phi, \bar\Psi)\right)$ also exists (by Lemma 2). Therefore, Donsker-Varadhan is applicable for $\tilde{g}$ and gives the following:

$$I(\Phi; \Psi) \geq \mathbb{E}_{\Phi, \Psi}\left[\tilde{g}(\Phi, \Psi)\right] - \log \mathbb{E}_{\Phi, \bar\Psi}\left[e^{\tilde{g}(\Phi, \bar\Psi)}\right] \tag{49}$$

$$= \lambda\,\mathbb{E}\left[\left(g(\Phi, \Psi) - \mathbb{E}_{\bar\Psi}\, g(\Phi, \bar\Psi)\right)^2\right] - \log \mathbb{E}_{\phi \sim \Phi}\,\mathbb{E}_{\bar\Psi}\exp\left\{\lambda\left(g(\phi, \bar\Psi) - \mathbb{E}_{\bar\Psi}\, g(\phi, \bar\Psi)\right)^2\right\} \tag{50}$$

$$\geq \lambda\,\mathbb{E}\left[\left(g(\Phi, \Psi) - \mathbb{E}_{\bar\Psi}\, g(\Phi, \bar\Psi)\right)^2\right] - \log\left(1 + 8\lambda\sigma^2\right) \qquad \text{(by Lemma 2).} \tag{51}$$

Picking $\lambda \to 1/(4\sigma^2)$, we get

$$I(\Phi; \Psi) \geq \frac{1}{4\sigma^2}\,\mathbb{E}\left[\left(g(\Phi, \Psi) - \mathbb{E}_{\bar\Psi}\, g(\Phi, \bar\Psi)\right)^2\right] - \log 3, \tag{52}$$

which proves the desired inequality. To prove the last part of the lemma, we just use the Markov's inequality and combine with this last result:

$$P\left(\left|g(\Phi, \Psi) - \mathbb{E}_{\bar\Psi}\, g(\Phi, \bar\Psi)\right| \geq \epsilon\right) = P\left(\left(g(\Phi, \Psi) - \mathbb{E}_{\bar\Psi}\, g(\Phi, \bar\Psi)\right)^2 \geq \epsilon^2\right) \tag{53}$$

$$\leq \frac{\mathbb{E}\left(g(\Phi, \Psi) - \mathbb{E}_{\bar\Psi}\, g(\Phi, \bar\Psi)\right)^2}{\epsilon^2} \tag{54}$$

$$\leq \frac{4\sigma^2(I(\Phi; \Psi) + \log 3)}{\epsilon^2}. \tag{55}$$

## A.2 Proof of Thm. 2.2

Let us fix a value of $U$. Conditioning on $U = u$ keeps the distribution of $W$ and $S$ intact, as $U$ is independent of $R$ and $S$. Let us set $\Phi = W$, $\Psi = S_u$, and

$$g(w, s_u) = \frac{1}{m}\sum_{i=1}^{m}\left(\ell(w, s_{u_i}) - \mathbb{E}_{Z' \sim \mathcal{D}}\,\ell(w, Z')\right). \tag{56}$$

Note that for each value of $w$, the random variable $g(w, \bar{S}_u)$ is $\frac{\sigma}{\sqrt{m}}$-subgaussian, as it is a sum of $m$ i.i.d. $\sigma$-subgaussian random variables. Furthermore, $\forall w$, $\mathbb{E}\, g(w, \bar{S}_u) = 0$. These two statements together and Lemma 3 imply that $g(W, \bar{S}_u)$ is also $\frac{\sigma}{\sqrt{m}}$-subgaussian. Therefore, with these choices of $\Phi$, $\Psi$, and $g$, Lemma 1 gives that

$$\left| \mathbb{E}_{S,R} \left[ \frac{1}{m} \sum_{i=1}^{m} \ell(W, S_{u_i}) - \mathbb{E}_{Z' \sim \mathcal{D}}\, \ell(W, Z') \right] \right| \leq \sqrt{\frac{2\sigma^2}{m} I(W; S_u)}. \tag{57}$$

Taking expectation over $u$ on both sides, then swapping the order between expectation over $u$ and absolute value (using Jensen's inequality), we get

$$\left| \mathbb{E}_{S,R,u \sim U} \left[ \frac{1}{m} \sum_{i=1}^{m} \ell(W, S_{u_i}) - \mathbb{E}_{Z' \sim \mathcal{D}}\, \ell(W, Z') \right] \right| \leq \mathbb{E}_{u \sim U} \sqrt{\frac{2\sigma^2}{m} I(W; S_u)}. \tag{58}$$

This proves the first part of the theorem as the left-hand side is equal to the absolute value of the expected generalization gap, $|\mathbb{E}_{S,R} [\mathcal{L}(A, S, R) - \mathcal{L}_{\mathrm{emp}}(A, S, R)]|$.

The second part of Lemma 1 gives that

$$\mathbb{E}_{S,R} \left( \frac{1}{m} \sum_{i=1}^{m} \ell(W, S_{u_i}) - \mathbb{E}_{Z' \sim \mathcal{D}}\, \ell(W, Z') \right)^2 \leq \frac{4\sigma^2}{m} \left( I(W; S_u) + \log 3 \right). \tag{59}$$

When $u = [n]$, (59) becomes

$$\mathbb{E}_{S,R} \left( \mathcal{L}(A, S, R) - \mathcal{L}_{\mathrm{emp}}(A, S, R) \right)^2 \leq \frac{4\sigma^2}{n} \left( I(W; S) + \log 3 \right), \tag{60}$$

proving the second part of the theorem.

Note that in case of $m = 1$ and $u = \{i\}$, (59) becomes

$$\mathbb{E}_{S,R} \left( \ell(W, Z_i) - \mathbb{E}_{Z' \sim \mathcal{D}}\, \ell(W, Z') \right)^2 \leq 4\sigma^2 \left( I(W; Z_i) + \log 3 \right). \tag{61}$$

Unfortunately, this result is not useful for bounding $\mathbb{E}_{S,R} \left( \mathcal{L}(A, S, R) - \mathcal{L}_{\mathrm{emp}}(A, S, R) \right)^2$, as for large $n$ the $\log 3$ term will likely dominate over $I(W; Z_i)$.

### A.3 Proof of Proposition 1

Before we prove Proposition 1, we establish two useful lemmas that will be helpful also in the proofs of Proposition 2, Proposition 3, Thm. 2.3, Thm. A.1 and Thm. 3.2.

**Lemma 4.** *Let $\Psi = (\Psi_1, \dots, \Psi_n)$ be a collection of $n$ independent random variables and $\Phi$ be another random variable defined on the same probability space. Then*

$$\forall i \in [n],\ I(\Phi; \Psi_i) \leq I(\Phi; \Psi_i \mid \Psi_{-i}), \tag{62}$$

*and*

$$I(\Phi; \Psi) \leq \sum_{i=1}^{n} I(\Phi; \Psi_i \mid \Psi_{-i}). \tag{63}$$

*Proof.* First, for all $i \in [n]$,

$$
\begin{aligned}
I(\Phi; \Psi_i \mid \Psi_{-i}) &= I(\Phi; \Psi_i) - I(\Psi_i; \Psi_{-i}) + I(\Psi_i; \Psi_{-i} \mid \Phi) && \text{(chain rule of MI)} && (64)\\
&= I(\Phi; \Psi_i) + I(\Psi_i; \Psi_{-i} \mid \Phi) && (\Psi_i \perp\!\!\!\perp \Psi_{-i}) && (65)\\
&\geq I(\Phi; \Psi_i) && \text{(nonnegativity of MI).} && (66)
\end{aligned}
$$

Second,

$$I(\Phi; \Psi) = \sum_{i=1}^{n} I(\Phi; \Psi_i \mid \Psi_{<i}) \tag{67}$$

$$= \sum_{i=1}^{n} \left( I(\Phi; \Psi_i \mid \Psi_{<i}, \Psi_{>i}) + I(\Psi_i; \Psi_{>i} \mid \Psi_{<i}) - I(\Psi_i; \Psi_{>i} \mid \Psi_{<i}, \Phi) \right) \tag{68}$$

$$= \sum_{i=1}^{n} \left( I(\Phi; \Psi_i \mid \Psi_{<i}, \Psi_{>i}) - I(\Psi_i; \Psi_{>i} \mid \Psi_{<i}, \Phi) \right) \tag{69}$$

$$\leq \sum_{i=1}^{n} I(\Phi; \Psi_i \mid \Psi_{<i}, \Psi_{>i}) \tag{70}$$

$$= \sum_{i=1}^{n} I(\Phi; \Psi_i \mid \Psi_{-i}). \tag{71}$$

The first two equalities above use the chain rule of mutual information, while the third one uses the independence of $\Psi_1, \ldots, \Psi_n$. The inequality of the fourth line relies on the nonnegativity of mutual information. $\qquad\square$

The quantity $\sum_{i=1}^{n} I(\Phi; \Psi_i \mid \Psi_{-i})$ is also known as erasure information and is denoted by $I^-(\Phi; \Psi)$ [34].

**Lemma 5.** *Let $S = (Z_1, \ldots, Z_n)$ be a collection of $n$ independent random variables and $\Phi$ be an arbitrary random variable defined on the same probability space. Then for any subset $u' \subseteq \{1, 2, \ldots, n\}$ of size $m + 1$ the following holds:*

$$I(\Phi; S_{u'}) \geq \frac{1}{m} \sum_{k \in u'} I(\Phi; S_{u' \setminus \{k\}}). \tag{72}$$

*Proof.*

$$(m+1)I(\Phi; S_{u'}) = \sum_{k \in u'} I(\Phi; S_{u' \setminus \{k\}}) + \sum_{k \in u'} I(\Phi; Z_k \mid S_{u' \setminus \{k\}}) \quad \text{(chain-rule of MI)} \tag{73}$$

$$\geq \sum_{k \in u'} I(\Phi; S_{u' \setminus \{k\}}) + I(\Phi; S_{u'}) \qquad \text{(second part of Lemma 4).}$$

$$\tag{74}$$

$$\square$$

**Proof of Proposition 1.** By Lemma 5 with $\Phi = W$, for any subset $u'$ of size $m + 1$ the following holds:

$$I(W; S_{u'}) \geq \frac{1}{m} \sum_{k \in u'} I(W; S_{u' \setminus \{k\}}). \tag{75}$$

Therefore,

$$\phi\left( \frac{1}{m+1} I(W; S_{u'}) \right) \geq \phi\left( \frac{1}{m(m+1)} \sum_{k \in u'} I(W; S_{u' \setminus \{k\}}) \right) \tag{76}$$

$$\geq \frac{1}{m+1} \sum_{k \in u'} \phi\left( \frac{1}{m} I(W; S_{u' \setminus \{k\}}) \right). \quad \text{(by Jensen's inequality)} \tag{77}$$

Taking expectation over $u'$ on both sides, we have hat

$$\mathbb{E}_{U'} \phi\left( \frac{1}{m+1} I(W; S_{u'}) \right) \geq \mathbb{E}_{U'} \left[ \frac{1}{m+1} \sum_{k \in u'} \phi\left( \frac{1}{m} I(W; S_{u' \setminus \{k\}}) \right) \right] \tag{78}$$

$$= \sum_{u} \alpha_u \phi\left( \frac{1}{m} I(W; S_u) \right). \tag{79}$$

For each subset $u$ of size $m$, the coefficient $\alpha_u$ is equal to

$$\alpha_u = \frac{1}{\binom{n}{m+1}} \cdot \frac{1}{m+1} \cdot (n-m) = \frac{1}{\binom{n}{m}}. \tag{80}$$

Therefore

$$\sum_u \alpha_u \phi\left(\frac{1}{m}I(W;S_u)\right) = \mathbb{E}_U\,\phi\left(\frac{1}{m}I(W;S_u)\right). \tag{81}$$

$\square$

## A.4  Proof of Thm. 2.3

Using Lemma 4 with $\Phi = W$ and $\Psi = S$, we get that

$$I(W;Z_i) \leq I(W;Z_i \mid Z_{-i}) \quad \text{and} \quad I(W;S) \leq \sum_{i=1} I(W;Z_i \mid Z_{-i}). \tag{82}$$

Plugging these upper bounds into Thm. 2.2 completes the proof.

## A.5  Proof of Thm. 2.6

Let us condition on $U = u$ and $\tilde{Z} = \tilde{z}$. Let $\Phi = A(\tilde{z}_S, R)$, $\Psi = S_u$, and

$$g(\phi, \psi) = \frac{1}{m} \sum_{i=1}^m \left(\ell(\phi, (\tilde{z}_u)_{i,\psi_i}) - \ell(\phi, (\tilde{z}_u)_{i,\mathrm{neg}(\psi_i)})\right). \tag{83}$$

Note that by our assumption, for any $w \in \mathcal{W}$ each summand of $g(w, \bar{S}_u)$ is in $[-1, +1]$, hence is a 1-subgaussian random variable. Furthermore, each of these summands has zero mean. As the average of $m$ independent and zero-mean 1-subgaussian variables is $\frac{1}{\sqrt{m}}$-subgaussian, then $g(w, \bar{S}_u)$ is $\frac{1}{\sqrt{m}}$-subgaussian for each $w \in \mathcal{W}$. Additionally, $\forall w \in \mathcal{W}$, $\mathbb{E}_{\bar{S}}\, g(w, \bar{S}_u) = 0$. Therefore, by Lemma 3, $g(\Phi, \bar{S}_u)$ is also $\frac{1}{\sqrt{m}}$-subgaussian. With these choices of $\Phi, \Psi$, and $g(\phi, \psi)$, we use Lemma 1. To avoid notational clutter we will denote the random variable $A(\tilde{z}_S, R)$ by $W$, hiding its dependence on $\tilde{z}, R, S$.[1] First,

$$\mathbb{E}_{S,R}\, g(W, S_u) = \mathbb{E}_{S,R}\left[\frac{1}{m} \sum_{i=1}^m \left(\ell(W, (\tilde{z}_u)_{i,(S_u)_i}) - \ell(W, (\tilde{z}_u)_{i,\mathrm{neg}((S_u)_i)})\right)\right]. \tag{84}$$

Second,

$$\mathbb{E}_{S,R,\bar{S}}\left[g(W, \bar{S}_u)\right] = 0. \tag{85}$$

Therefore, Lemma 1 gives

$$\left|\mathbb{E}_{S,R}\left[\frac{1}{m} \sum_{i=1}^m \left(\ell(W, (\tilde{z}_u)_{i,(S_u)_i}) - \ell(W, (\tilde{z}_u)_{i,\mathrm{neg}((S_u)_i)})\right)\right]\right| \leq \sqrt{\frac{2}{m}I(W;S_u)}. \tag{86}$$

Taking expectation over $u$ on both sides and using Jensen's inequality to switch the order of absolute value and expectation of $u$, we get

$$\left|\mathbb{E}_{S,R,u\sim U}\left[\frac{1}{m} \sum_{i=1}^m \left(\ell(W, (\tilde{z}_u)_{i,(S_u)_i}) - \ell(W, (\tilde{z}_u)_{i,\mathrm{neg}((S_u)_i)})\right)\right]\right| \leq \mathbb{E}_{u\sim U}\sqrt{\frac{2}{m}I(W;S_u)}, \tag{87}$$

which reduces to

$$\left|\mathbb{E}_{S,R}\left[\frac{1}{n} \sum_{i=1}^n \left(\ell(W, \tilde{z}_{i,S_i}) - \ell(W, \tilde{z}_{i,\mathrm{neg}(S_i)})\right)\right]\right| \leq \mathbb{E}_{u\sim U}\sqrt{\frac{2}{m}I(W;S_u)}. \tag{88}$$

---

[1] Expressions like $\mathbb{E}_{\tilde{z}\sim\tilde{Z}}\, W$ will mean $\mathbb{E}_{\tilde{z}\sim\tilde{Z}}\, A(\tilde{z}_S, R)$.

This can be seen as bounding the expected generalization gap for a fixed $\tilde{z}$. Taking expectation over $\tilde{z}$ on both sides, and then using Jensen's inequality to switch the order of absolute value and expectation of $\tilde{z}$, we get

$$\left| \mathbb{E}_{\tilde{z} \sim \tilde{Z}, S, R} \left[ \frac{1}{n} \sum_{i=1}^{n} \left( \ell(W, \tilde{z}_{i,S_i}) - \ell(W, \tilde{z}_{i,\text{neg}(S_i)}) \right) \right] \right| \leq \mathbb{E}_{\tilde{z} \sim \tilde{Z}, u \sim U} \sqrt{\frac{2}{m} I(W; S_u)} \quad (89)$$

Finally, noticing that left-hand side is equal to the absolute value of the expected generalization gap, $\left| \mathbb{E}_{\tilde{Z}, S, R} \left[ \mathcal{L}(A, \tilde{Z}_S, R) - \mathcal{L}_{\text{emp}}(A, \tilde{Z}_S, R) \right] \right|$, completes the proof of the first part of this theorem.

When $u = [n]$, applying the second part of Lemma 1 gives

$$\mathbb{E}_{S,R} \left( \frac{1}{n} \sum_{i=1}^{n} \left( \ell(W, \tilde{z}_{i,S_i}) - \ell(W, \tilde{z}_{i,\text{neg}(S_i)}) \right) \right)^2 \leq \frac{4}{n} (I(W; S) + \log 3). \quad (90)$$

Taking expectation over $\tilde{z}$, we get

$$\underbrace{\mathbb{E}_{\tilde{z} \sim \tilde{Z}, S, R} \left( \frac{1}{n} \sum_{i=1}^{n} \left( \ell(W, \tilde{z}_{i,S_i}) - \ell(W, \tilde{z}_{i,\text{neg}(S_i)}) \right) \right)^2}_{B} \leq \frac{4}{n} \mathbb{E}_{\tilde{z} \sim \tilde{Z}} (I(W; S) + \log 3). \quad (91)$$

Continuing,

$$\mathbb{E}_{\tilde{Z}, S, R} \left( \mathcal{L}(A, \tilde{Z}_S, R) - \mathcal{L}_{\text{emp}}(A, \tilde{Z}_S, R) \right)^2 = \mathbb{E}_{\tilde{z} \sim \tilde{Z}, S, R} \left( \frac{1}{n} \sum_{i=1}^{n} \ell(W, \tilde{z}_{i,S_i}) - \mathbb{E}_{Z' \sim \mathcal{D}} \ell(W, Z') \right)^2 \quad (92)$$

$$\leq 2B + 2 \mathbb{E}_{\tilde{z} \sim \tilde{Z}, S, R} \left( \frac{1}{n} \sum_{i=1}^{n} \ell(W, \tilde{z}_{i,\text{neg}(S_i)}) - \mathbb{E}_{Z' \sim \mathcal{D}} \ell(W, Z') \right)^2 \quad (93)$$

$$= 2B + 2 \mathbb{E}_{\tilde{z} \sim \tilde{Z}, S, R} \left( \frac{1}{n} \sum_{i=1}^{n} \left( \ell(W, \tilde{z}_{i,\text{neg}(S_i)}) - \mathbb{E}_{Z' \sim \mathcal{D}} \ell(W, Z') \right) \right)^2 \quad (94)$$

$$= 2B + 2 \mathbb{E}_{\tilde{Z}_S, \tilde{Z}_{\text{neg}(S)}, R} \left( \frac{1}{n} \sum_{i=1}^{n} \left( \ell(A(\tilde{Z}_S, R), (\tilde{Z}_{\text{neg}(S)})_i) - \mathbb{E}_{Z' \sim \mathcal{D}} \ell(A(\tilde{Z}_S, R), Z') \right) \right)^2 \quad (95)$$

$$= 2B + 2 \mathbb{E}_{\tilde{Z}_S, R} \mathbb{E}_{\tilde{Z}_{\text{neg}(S)} | \tilde{Z}_S, R} \left( \frac{1}{n} \sum_{i=1}^{n} \left( \ell(A(\tilde{Z}_S, R), (\tilde{Z}_{\text{neg}(S)})_i) - \mathbb{E}_{Z' \sim \mathcal{D}} \ell(A(\tilde{Z}_S, R), Z') \right) \right)^2. \quad (96)$$

Note that as $\tilde{Z}_{\text{neg}(S)}$ is independent of $(\tilde{Z}_S, R)$, conditioning on $(\tilde{Z}_S, R)$ does not change its distribution, implying that its components stay independent of each other. For each fixed values $\tilde{Z}_S = z$ and $R = r$, the inner part of the outer expectation in (96) becomes

$$\mathbb{E}_{\tilde{Z}_{\text{neg}(S)}} \left( \frac{1}{n} \sum_{i=1}^{n} \left( \ell(A(z, r), (\tilde{Z}_{\text{neg}(S)})_i) - \mathbb{E}_{Z' \sim \mathcal{D}} \ell(A(z, r), Z') \right) \right)^2, \quad (97)$$

which is equal to

$$\mathbb{E}_{Z'_1, Z'_2, \dots, Z'_n} \left( \frac{1}{n} \sum_{i=1}^{n} \left( \ell(A(z, r), Z'_i) - \mathbb{E}_{Z'_i} \ell(A(z, r), Z'_i) \right) \right)^2, \quad (98)$$

where $Z'_1, \dots, Z'_n$ are $n$ i.i.d. samples from $\mathcal{D}$. The expression in (98) is simply the variance of the average of $n$ i.i.d $[0, 1]$-bounded random variables. Hence, it can be bounded by $1/(4n)$. Connecting

this result with (96), we get

$$\mathbb{E}_{\tilde{Z},S,R}\left(\mathcal{L}(A,\tilde{Z}_S,R) - \mathcal{L}_{\mathrm{emp}}(A,\tilde{Z}_S,R)\right)^2 \le 2B + \frac{1}{2n} \tag{99}$$

$$\le 2\,\mathbb{E}_{\tilde{z}\sim\tilde{Z}}\left[\frac{4}{n}(I(W;S) + \log 3)\right] + \frac{1}{2n} \tag{100}$$

$$\le \mathbb{E}_{\tilde{z}\sim\tilde{Z}}\left[\frac{8}{n}(I(W;S) + 2)\right]. \tag{101}$$

## A.6   Proof of Proposition 2

The proof follows that of Proposition 1, with the only difference that $W$ is replaced with $A(\tilde{z}_S, R)$.

## A.7   Thm. A.1

**Theorem A.1.** *If $\ell(w,z) \in [0,1], \forall w \in \mathcal{W}, z \in \mathcal{Z}$, then*

$$\left|\mathbb{E}_{\tilde{Z},S,R}\left[\mathcal{L}(A,\tilde{Z}_S,R) - \mathcal{L}_{\mathrm{emp}}(A,\tilde{Z}_S,R)\right]\right| \le \mathbb{E}_{\tilde{z}\sim\tilde{Z}}\left[\frac{1}{n}\sum_{i=1}^{n}\sqrt{2I(A(\tilde{z}_S,R);S_i \mid S_{-i})}\right], \tag{102}$$

*and*

$$\mathbb{E}_{\tilde{Z},S,R}\left(\mathcal{L}(A,\tilde{Z}_S,R) - \mathcal{L}_{\mathrm{emp}}(A,\tilde{Z}_S,R)\right)^2 \le \frac{8}{n}\left(\mathbb{E}_{\tilde{z}\sim\tilde{Z}}\left[\sum_{i=1}^{n}I(A(\tilde{z}_S,R);S_i \mid S_{-i})\right] + 2\right). \tag{103}$$

*Proof.* For a fixed $\tilde{z}$, using Lemma 4 with $\Phi = A(\tilde{z}_S, R)$ and $\Psi = S$, we get that

$$I(A(\tilde{z}_S,R);S_i) \le I(A(\tilde{z}_S,R);S_i \mid S_{-i}), \tag{104}$$

and

$$I(A(\tilde{z}_S,R);S) \le \sum_{i=1}^{n}I(A(\tilde{z}_S,R);S_i \mid S_{-i}). \tag{105}$$

Using these upper bounds in Thm. 2.6 proves the theorem. □

## A.8   Proof of Thm. 3.1

Let us condition on $U = u$ and $\tilde{Z} = \tilde{z}$. To simplify the notation we will use $F$ as a shorthand for $f(\tilde{z}_S, \tilde{x}, R)$, the predictions on the all $n$ pairs after training on $\tilde{z}_S$ with randomness $R$. Expressions like $\mathbb{E}_{\tilde{z}\sim\tilde{Z}} F$ will mean $\mathbb{E}_{\tilde{z}\sim\tilde{Z}} f(\tilde{z}_S, \tilde{x}, R)$. Furthermore, when $\widehat{y}$ and $y$ are collection of predictions and labels, we will use $\ell(\widehat{y}, y)$ to denote the average loss.

We are going to use Lemma 1 with $\Phi = F_u$, $\Psi = S_u$, and

$$g(\phi, \psi) = \ell(\phi_\psi, (\tilde{y}_u)_\psi) - \ell(\phi_{\mathrm{neg}(\psi)}, (\tilde{y}_u)_{\mathrm{neg}(\psi)}) \tag{106}$$

$$= \frac{1}{m}\left(\sum_{i=1}^{m}\ell(\phi_{i,\psi_i}, (\tilde{y}_u)_{i,\psi_i}) - \ell(\phi_{i,\mathrm{neg}(\psi)_i}, (\tilde{y}_u)_{i,\mathrm{neg}(\psi)_i})\right). \tag{107}$$

The function $g(\phi, \psi)$ computes the generalization gap measured on pairs of the examples specified by subset $u$, assuming that predictions are given by $\phi$ and the training/test set split is given by $\psi$. Note that by our assumption, for any $\phi$ each summand of $g(\phi, \bar{S}_u)$ is a 1-subgaussian random variable. Furthermore, each of these summands has zero mean. As the average of $m$ independent and zero-mean 1-subgaussian variables is $\frac{1}{\sqrt{m}}$-subgaussian, then $g(\phi, \bar{S}_u)$ is $\frac{1}{\sqrt{m}}$-subgaussian for each possible $\phi$. Additionally, $\forall \phi \in \mathcal{K}^{m\times 2}$, $\mathbb{E}_{\bar{S}}\, g(\phi, \bar{S}_u) = 0$. By Lemma 3, $g(F_u, \bar{S}_u)$ is also $\frac{1}{\sqrt{m}}$-subgaussian. Hence, these choices of $\Phi$, $\Psi$, and $g(\phi, \psi)$ satisfy the assumptions of Lemma 1. We have that

$$\mathbb{E}_{S,R}\, g(F_u, S_u) = \mathbb{E}_{S,R}\left[\ell((F_u)_{S_u}, (\tilde{y}_u)_{S_u}) - \ell((F_u)_{\mathrm{neg}(S)_u}, (\tilde{y}_u)_{\mathrm{neg}(S)_u})\right], \tag{108}$$

and

$$\mathbb{E}_{F_u, \bar{S}_u} \left[ g(F_u, \bar{S}_u) \right] = 0. \tag{109}$$

Therefore, the first part of Lemma 1 gives

$$\left| \mathbb{E}_{S,F} \left[ \ell((F_u)_{S_u}, (\tilde{y}_u)_{S_u}) - \ell((F_u)_{\mathrm{neg}(S)_u}, (\tilde{y}_u)_{\mathrm{neg}(S)_u}) \right] \right| \leq \sqrt{\frac{2}{m} I(F_u; S_u)}. \tag{110}$$

Taking expectation over $u$ on both sides, and then using Jensen's inequality to swap the order of absolute value and expectation of $u$, we get

$$\left| \mathbb{E}_{S,R,u \sim U} \left[ \ell((F_u)_{S_u}, (\tilde{y}_u)_{S_u}) - \ell((F_u)_{\mathrm{neg}(S)_u}, (\tilde{y}_u)_{\mathrm{neg}(S)_u}) \right] \right| \leq \mathbb{E}_{u \sim U} \sqrt{\frac{2}{m} I(F_u; S_u)}, \tag{111}$$

which reduces to

$$\left| \mathbb{E}_{S,R} \left[ \ell(F_S, \tilde{y}_S) - \ell(F_{\mathrm{neg}(S)}, \tilde{y}_{\mathrm{neg}(S)}) \right] \right| \leq \mathbb{E}_{u \sim U} \sqrt{\frac{2}{m} I(F_u; S_u)}. \tag{112}$$

This can be seen as bounding the expected generalization gap for a fixed $\tilde{z}$. Taking expectation over $\tilde{z}$ on both sides and using Jensen's inequality to switch the order of absolute value and expectation of $\tilde{z}$, we get

$$\left| \mathbb{E}_{\tilde{z} \sim \tilde{Z}, S, R} \left[ \ell(F_S, \tilde{y}_S) - \ell(F_{\mathrm{neg}(S)}, \tilde{y}_{\mathrm{neg}(S)}) \right] \right| \leq \mathbb{E}_{\tilde{z} \sim \tilde{Z}, u \sim U} \sqrt{\frac{2\sigma^2}{m} I(F_u; S_u)}. \tag{113}$$

Noticing that the left-hand side is equal to the absolute value of the expected generalization gap, $\mathbb{E}_{\tilde{Z}, R, S} \left[ \mathcal{L}(f, \tilde{Z}_S, R) - \mathcal{L}_{\mathrm{emp}}(f, \tilde{Z}_S, R) \right]$, completes the proof of the first part of the theorem.

When $u = [n]$, applying the second part of Lemma 1 gives

$$\mathbb{E}_{S,R} \left( \ell(F_S, \tilde{y}_S) - \ell(F_{\mathrm{neg}(S)}, \tilde{y}_{\mathrm{neg}(S)}) \right)^2 \leq \frac{4}{n} (I(F; S) + \log 3). \tag{114}$$

Taking expectation over $\tilde{z}$, we get

$$\mathbb{E}_{\tilde{z} \sim \tilde{Z}, S, R} \left( \ell(F_S, \tilde{y}_S) - \ell(F_{\mathrm{neg}(S)}, \tilde{y}_{\mathrm{neg}(S)}) \right)^2 \leq \mathbb{E}_{\tilde{z} \sim \tilde{Z}} \left[ \frac{4}{n} (I(F; S) + \log 3) \right]. \tag{115}$$

Continuing,

$$\mathbb{E}_{\tilde{Z}, R, S} \left( \mathcal{L}(f, \tilde{Z}_S, R) - \mathcal{L}_{\mathrm{emp}}(f, \tilde{Z}_S, R) \right)^2 = \underbrace{\mathbb{E}_{\tilde{z} \sim \tilde{Z}, S, R} \left( \ell(F_S, \tilde{y}_S) - \mathbb{E}_{Z' \sim \mathcal{D}} \ell(f(\tilde{z}_S, X', R), Y') \right)^2}_{} \tag{116}$$

$$\leq 2B + 2 \mathbb{E}_{\tilde{z} \sim \tilde{Z}, S, R} \left( \ell(F_{\mathrm{neg}(S)}, \tilde{y}_{\mathrm{neg}(S)}) - \mathbb{E}_{Z' \sim \mathcal{D}} \ell(f(\tilde{z}_S, X', R), Y') \right)^2 \tag{117}$$

$$= 2B + 2 \mathbb{E}_{\tilde{z} \sim \tilde{Z}, S, R} \left( \frac{1}{n} \sum_{i=1}^{n} \left( \ell(F_{i,\mathrm{neg}(S)_i}, \tilde{y}_{i,\mathrm{neg}(S)_i}) - \mathbb{E}_{Z' \sim \mathcal{D}} \ell(f(\tilde{z}_S, X', R), Y') \right) \right)^2 \tag{118}$$

$$= 2B + 2 \mathbb{E}_{\tilde{Z}_S, R, \tilde{Z}_{\mathrm{neg}(S)}} \left( \frac{1}{n} \sum_{i=1}^{n} \left( \ell(f(\tilde{Z}_S, \tilde{X}_{\mathrm{neg}(S)}, R)_i, (\tilde{Y}_{\mathrm{neg}(S)})_i) - \mathbb{E}_{Z' \sim \mathcal{D}} \ell(f(\tilde{Z}_S, X', R), Y') \right) \right)^2 \tag{119}$$

$$= 2B + 2 \mathbb{E}_{\tilde{Z}_S, R} \mathbb{E}_{\tilde{Z}_{\mathrm{neg}(S)} | \tilde{Z}_S, R} \left( \frac{1}{n} \sum_{i=1}^{n} \left( \ell(f(\tilde{Z}_S, \tilde{X}_{\mathrm{neg}(S)}, R)_i, (\tilde{Y}_{\mathrm{neg}(S)})_i) - \mathbb{E}_{Z' \sim \mathcal{D}} \ell(f(\tilde{Z}_S, X', R), Y') \right) \right)^2. \tag{120}$$

Note that as $\tilde{Z}_{\mathrm{neg}(S)}$ is independent of $(\tilde{Z}_S, R)$, conditioning on $(\tilde{Z}_S, R)$ does not change its distribution, implying that its components stay independent of each other. For each fixed values $\tilde{Z}_S = z$ and $R = r$, the inner part of the expectation in (120) becomes

$$\mathbb{E}_{\tilde{Z}_{\mathrm{neg}(S)}} \left( \frac{1}{n} \sum_{i=1}^{n} \left( \ell(f(z, \tilde{X}_{\mathrm{neg}(S)}, r)_i, (\tilde{Y}_{\mathrm{neg}(S)})_i) - \mathbb{E}_{Z' \sim \mathcal{D}} \ell(f(z, X', r), Y') \right) \right)^2, \tag{121}$$

which is equal to

$$\mathbb{E}_{Z_1', Z_2', \ldots, Z_n'} \left( \frac{1}{n} \sum_{i=1}^{n} \left( \ell(f(z, X_i', Y_i') - \mathbb{E}_{Z_i'} \ell(f(z, X_i', r), Y_i')) \right) \right)^2, \tag{122}$$

where $Z_1', \ldots, Z_n'$ are $n$ i.i.d. samples from $\mathcal{D}$. The expression in (122) is simply the variance of the average of $n$ i.i.d $[0-1]$-bounded random variables. Hence, it can be bounded by $1/(4n)$. Connecting this result with (120), we get

$$\mathbb{E}_{\tilde{Z}, R, S} \left( \mathcal{L}(f, \tilde{Z}_S, R) - \mathcal{L}_{\text{emp}}(f, \tilde{Z}_S, R) \right)^2 \leq 2B + \frac{1}{2n} \tag{123}$$

$$\leq 2\, \mathbb{E}_{\tilde{z} \sim \tilde{Z}} \left[ \frac{4}{n} (I(F; S) + \log 3) \right] + \frac{1}{2n} \tag{124}$$

$$\leq \mathbb{E}_{\tilde{z} \sim \tilde{Z}} \left[ \frac{8}{n} (I(F; S) + 2) \right]. \tag{125}$$

### A.9   Proof of Proposition 3

The proof closely follows that of Proposition 1. The only difference is that $f(\tilde{z}_S, \tilde{x}_u, R)$ depends on $u$, while $W$ does not.

If we fix a subset $u'$ of size $m + 1$, set $\Phi = f(\tilde{z}_S, \tilde{x}_{u'}, R))$, and use Lemma 5, we get

$$I(f(\tilde{z}_S, \tilde{x}_{u'}, R); S_{u'}) \geq \frac{1}{m} \sum_{k \in u'} I(f(\tilde{z}_S, \tilde{x}_{u'}, R); S_{u' \setminus \{k\}}) \tag{126}$$

$$\geq \frac{1}{m} \sum_{k \in u'} I(f(\tilde{z}_S, \tilde{x}_{u' \setminus \{k\}}, R); S_{u' \setminus \{k\}}). \quad \text{(removing predictions on pair } k \text{ can not increase MI)} \tag{127}$$

Therefore,

$$\phi \left( \frac{1}{m+1} I(f(\tilde{z}_S, \tilde{x}_{u'}, R); S_{u'}) \right) \geq \phi \left( \frac{1}{m(m+1)} \sum_{k \in u'} I(f(\tilde{z}_S, \tilde{x}_{u' \setminus \{k\}}, R); S_{u' \setminus \{k\}}) \right) \tag{128}$$

$$\geq \frac{1}{m+1} \sum_{k \in u'} \phi \left( \frac{1}{m} I(f(\tilde{z}_S, \tilde{x}_{u' \setminus \{k\}}, R); S_{u' \setminus \{k\}}) \right) \quad \text{(by Jensen's inequality)} \tag{129}$$

Taking expectation over $U'$ on both sides, we have

$$\mathbb{E}_{U'} \phi \left( \frac{1}{m+1} I(f(\tilde{z}_S, \tilde{x}_{u'}, R); S_{u'}) \right) \geq \mathbb{E}_{U'} \left[ \frac{1}{m+1} \sum_{k \in u'} \phi \left( \frac{1}{m} I(f(\tilde{z}_S, \tilde{x}_{u' \setminus \{k\}}, R); S_{u' \setminus \{k\}}) \right) \right] \tag{130}$$

$$= \sum_{u} \alpha_u \phi \left( \frac{1}{m} I(f(\tilde{z}_S, \tilde{x}_u, R); S_u) \right). \tag{131}$$

For each subset $u$ of size $m$, the coefficient $\alpha_u$ is equal to

$$\alpha_u = \frac{1}{\binom{n}{m+1}} \cdot \frac{1}{m+1} \cdot (n - m) = \frac{1}{\binom{n}{m}}. \tag{132}$$

Therefore

$$\sum_{u} \alpha_u \phi \left( \frac{1}{m} I(f(\tilde{z}_S, \tilde{x}_u, R); S_u) \right) = \mathbb{E}_U \, \phi \left( \frac{1}{m} I(f(\tilde{z}_S, \tilde{x}_u, R); S_u) \right). \tag{133}$$

## A.10 Proof of Thm. 3.2

Let us fix $\tilde{z}$. Setting $\Phi = f(\tilde{z}_S, \tilde{x}_i, R)$, $\Psi = S$, and using the first part of Lemma 4, we get that

$$I(f(\tilde{z}_S, \tilde{x}_i, R); S_i) \leq I(f(\tilde{z}_S, \tilde{x}_i, R); S_i \mid S_{-i}). \tag{134}$$

Next, setting $\Phi = f(\tilde{z}_S, \tilde{x}, R)$, $\Psi = S$, and using the second part of Lemma 4, we get that

$$I(f(\tilde{z}_S, \tilde{x}, R); S) \leq \sum_{i=1}^{n} I(f(\tilde{z}_S, \tilde{x}, R); S_i \mid S_{-i}). \tag{135}$$

Using these upper bounds in Thm. 3.1 proves this theorem.

## A.11 Proof of Thm. 4.1

Let $k$ denote the number of distinct values $f(\tilde{z}_S, \tilde{x}, R)$ can take by varying $S$ and $R$. Clearly, $k$ is not more than the growth function of $\mathcal{H}$ evaluated at $2n$. Applying the Sauer-Shelah lemma [29, 30], we get that

$$k \leq \sum_{i=0}^{d} \binom{2n}{i}. \tag{136}$$

The Sauer-Shelah lemma also states that if $2n > d + 1$ then

$$\sum_{i=0}^{d} \binom{2n}{i} \leq \left(\frac{2en}{d}\right)^{d}. \tag{137}$$

If $2n \leq d + 1$, one can upper bound $k$ by $2^{2n} \leq 2^{d+1}$. Therefore

$$k \leq \max\left\{2^{d+1}, \left(\frac{2en}{d}\right)^{d}\right\}. \tag{138}$$

Finally, as a $f(\tilde{z}_S, \tilde{x}, R)$ is a discrete variable with $k$ states,

$$f\text{-CMI}(f, \tilde{z}) \leq H(f(\tilde{z}_S, \tilde{x}, R)) \leq \log(k). \tag{139}$$

## A.12 Proof of Proposition 4

The proof below uses the independence of $S_1, \ldots, S_n$ and the convexity of KL divergence, once for the first and once for the second argument.

$$I(f(\tilde{z}_S, x, R); S_i \mid S_{-i}) = \text{KL}\left(f(\tilde{z}_S, x, R)|S \parallel f(\tilde{z}_S, x, R)|S_{-i}\right) \tag{140}$$

$$= \frac{1}{2}\text{KL}\left(f(\tilde{z}_{S^{i \leftarrow 0}}, x, R)|S_{-i} \parallel f(\tilde{z}_S, x, R)|S_{-i}\right) \tag{141}$$

$$+ \frac{1}{2}\text{KL}\left(f(\tilde{z}_{S^{i \leftarrow 1}}, x, R)|S_{-i} \parallel f(\tilde{z}_S, x, R)|S_{-i}\right) \tag{142}$$

$$= \frac{1}{2}\text{KL}\left(f(\tilde{z}_{S^{i \leftarrow 0}}, x, R)|S_{-i} \parallel \frac{1}{2}\left(f(\tilde{z}_{S^{i \leftarrow 0}}, x, R) + f(\tilde{z}_{S^{i \leftarrow 1}}, x, R)\right)|S_{-i}\right) \tag{143}$$

$$+ \frac{1}{2}\text{KL}\left(f(\tilde{z}_{S^{i \leftarrow 1}}, x, R)|S_{-i} \parallel \frac{1}{2}\left(f(\tilde{z}_{S^{i \leftarrow 0}}, x, R) + f(\tilde{z}_{S^{i \leftarrow 1}}, x, R)\right)|S_{-i}\right) \tag{144}$$

$$\leq \frac{1}{4}\text{KL}\left(f(\tilde{z}_{S^{i \leftarrow 0}}, x, R)|S_{-i} \parallel f(\tilde{z}_{S^{i \leftarrow 0}}, x, R)|S_{-i}\right) \tag{145}$$

$$+ \frac{1}{4}\text{KL}\left(f(\tilde{z}_{S^{i \leftarrow 0}}, x, R)|S_{-i} \parallel f(\tilde{z}_{S^{i \leftarrow 1}}, x, R)|S_{-i}\right) \tag{146}$$

$$+ \frac{1}{4}\text{KL}\left(f(\tilde{z}_{S^{i \leftarrow 1}}, x, R)|S_{-i} \parallel f(\tilde{z}_{S^{i \leftarrow 0}}, x, R)|S_{-i}\right) \tag{147}$$

$$+ \frac{1}{4}\text{KL}\left(f(\tilde{z}_{S^{i \leftarrow 1}}, x, R)|S_{-i} \parallel f(\tilde{z}_{S^{i \leftarrow 1}}, x, R)|S_{-i}\right) \tag{148}$$

$$= \frac{1}{4}\text{KL}\left(f(\tilde{z}_{S^{i \leftarrow 0}}, x, R)|S_{-i} \parallel f(\tilde{z}_{S^{i \leftarrow 1}}, x, R)|S_{-i}\right) \tag{149}$$

$$+ \frac{1}{4}\text{KL}\left(f(\tilde{z}_{S^{i \leftarrow 1}}, x, R)|S_{-i} \parallel f(\tilde{z}_{S^{i \leftarrow 0}}, x, R)|S_{-i}\right). \tag{150}$$

## A.13 Proof of Thm. 4.2

Given a deterministic algorithm $f$, we consider the algorithm that adds Gaussian noise to the predictions of $f$:

$$f_\sigma(z, x, R) = f(z, x) + \xi, \tag{151}$$

where $\xi \sim \mathcal{N}(0, \sigma^2 I_d)$. The function $f_\sigma$ is constructed in a way that the noise terms are independent for each possible combination of $z$ and $x$.[2]

First we relate the generalization gap of $f_\sigma$ to that of $f$:

$$\left| \mathbb{E}_{\tilde{Z}, R, S} \left[ \mathcal{L}(f_\sigma, \tilde{Z}_S, R) - \mathcal{L}_{\text{emp}}(f_\sigma, \tilde{Z}_S, R) \right] \right| \tag{152}$$

$$= \left| \mathbb{E}_{\tilde{Z}, R, S, Z' \sim \mathcal{D}} \left[ \ell(f_\sigma(\tilde{Z}_S, X', R), Y') - \frac{1}{n} \sum_{i=1}^{n} \ell(f_\sigma(\tilde{Z}_S, \tilde{X}_{i, S_i}, R), \tilde{Y}_{i, S_i}) \right] \right| \tag{153}$$

$$= \left| \mathbb{E}_{\tilde{Z}, R, S, Z' \sim \mathcal{D}} \left[ \ell(f(\tilde{Z}_S, X') + \xi', Y') - \frac{1}{n} \sum_{i=1}^{n} \ell(f(\tilde{Z}_S, \tilde{X}_{i, S_i}) + \xi_i, \tilde{Y}_{i, S_i}) \right] \right| \tag{154}$$

$$= \left| \mathbb{E}_{\tilde{Z}, R, S, Z' \sim \mathcal{D}} \left[ \ell(f(\tilde{Z}_S, X'), Y') + \Delta' - \frac{1}{n} \sum_{i=1}^{n} \left( \ell(f(\tilde{Z}_S, \tilde{X}_{i, S_i}), \tilde{Y}_{i, S_i}) + \Delta_i \right) \right] \right|, \tag{155}$$

where $\Delta' = \ell(f(\tilde{Z}_S, X') + \xi', Y') - \ell(f(\tilde{Z}_S, X'), Y')$ and $\Delta_i = \ell(f(\tilde{Z}_S, \tilde{X}_{i, S_i}) + \xi_i, \tilde{Y}_{i, S_i}) - \ell(f(\tilde{Z}_S, \tilde{X}_{i, S_i}), \tilde{Y}_{i, S_i})$. As $\ell(\hat{y}, y)$ is $\gamma$-Lipschitz in its first argument, $|\Delta'| \leq \gamma \|\xi'\|$ and $|\Delta_i| \leq \gamma \|\xi_i\|$. Connecting this to (155) we get

$$\left| \mathbb{E}_{\tilde{Z}, R, S} \left[ \mathcal{L}(f_\sigma, \tilde{Z}_S, R) - \mathcal{L}_{\text{emp}}(f_\sigma, \tilde{Z}_S, R) \right] \right| \geq \left| \mathbb{E}_{\tilde{Z}, R, S} \left[ \mathcal{L}(f, \tilde{Z}_S) - \mathcal{L}_{\text{emp}}(f, \tilde{Z}_S) \right] \right| \tag{156}$$

$$- \gamma \, \mathbb{E} \|\xi'\| - \frac{\gamma}{n} \sum_{i=1}^{n} \mathbb{E} \|\xi_i\| \tag{157}$$

$$= \left| \mathbb{E}_{\tilde{Z}, R, S} \left[ \mathcal{L}(f, \tilde{Z}_S) - \mathcal{L}_{\text{emp}}(f, \tilde{Z}_S) \right] \right| - 2\sqrt{d}\gamma\sigma. \tag{158}$$

Similarly, we relate the expected squared generalization gap of $f_\sigma$ to that of $f$:

$$\mathbb{E}_{\tilde{Z}, R, S} \left( \mathcal{L}(f_\sigma, \tilde{Z}_S, R) - \mathcal{L}_{\text{emp}}(f_\sigma, \tilde{Z}_S, R) \right)^2 \tag{159}$$

$$= \mathbb{E}_{\tilde{Z}, R, S} \left( \mathbb{E}_{Z' \sim \mathcal{D}} \left[ \ell(f(\tilde{Z}_S, X'), Y') + \Delta' \right] - \frac{1}{n} \sum_{i=1}^{n} \left( \ell(f(\tilde{Z}_S, \tilde{X}_{i, S_i}), \tilde{Y}_{i, S_i}) + \Delta_i \right) \right)^2 \tag{160}$$

$$= \mathbb{E}_{\tilde{Z}, R, S} \left( \mathcal{L}(f, \tilde{Z}_S) - \mathcal{L}_{\text{emp}}(f, \tilde{Z}_S) \right)^2 + \mathbb{E}_{\tilde{Z}, R, S} \left( \mathbb{E}_{Z' \sim \mathcal{D}} [\Delta'] - \frac{1}{n} \sum_{i=1}^{n} \Delta_i \right)^2 \tag{161}$$

$$+ 2 \mathbb{E}_{\tilde{Z}, R, S} \left[ \left( \mathcal{L}(f, \tilde{Z}_S) - \mathcal{L}_{\text{emp}}(f, \tilde{Z}_S) \right) \left( \mathbb{E}_{Z' \sim \mathcal{D}} [\Delta'] - \frac{1}{n} \sum_{i=1}^{n} \Delta_i \right) \right] \tag{162}$$

$$\geq \mathbb{E}_{\tilde{Z}, R, S} \left( \mathcal{L}(f, \tilde{Z}_S) - \mathcal{L}_{\text{emp}}(f, \tilde{Z}_S) \right)^2 \tag{163}$$

$$- 2 \mathbb{E}_{\tilde{Z}, R, S} \left[ \left| \mathcal{L}(f, \tilde{Z}_S) - \mathcal{L}_{\text{emp}}(f, \tilde{Z}_S) \right| \left| \mathbb{E}_{Z' \sim \mathcal{D}} [\Delta'] - \frac{1}{n} \sum_{i=1}^{n} \Delta_i \right| \right] \tag{164}$$

$$= \mathbb{E}_{\tilde{Z}, S} \left( \mathcal{L}(f, \tilde{Z}_S) - \mathcal{L}_{\text{emp}}(f, \tilde{Z}_S) \right)^2 \tag{165}$$

$$- 2 \mathbb{E}_{\tilde{Z}, S} \left[ \left| \mathcal{L}(f, \tilde{Z}_S) - \mathcal{L}_{\text{emp}}(f, \tilde{Z}_S) \right| \mathbb{E}_R \left[ \left| \mathbb{E}_{Z' \sim \mathcal{D}} [\Delta'] - \frac{1}{n} \sum_{i=1}^{n} \Delta_i \right| \right] \right]. \tag{166}$$

---

[2]This can be achieved by viewing $R$ as an infinite collection of independent Gaussian variables, one of which is selected for each possible combination of $z$ and $x$.

As

$$\mathbb{E}_R\left[\left|\mathbb{E}_{Z'\sim\mathcal{D}}[\Delta'] - \frac{1}{n}\sum_{i=1}^n \Delta_i\right|\right] \le \mathbb{E}_R\left[\mathbb{E}_{Z'\sim\mathcal{D}}|\Delta'|\right] + \frac{1}{n}\sum_{i=1}^n \mathbb{E}_R|\Delta_i| \tag{167}$$

$$\le \mathbb{E}_R\left[\mathbb{E}_{Z'\sim\mathcal{D}}\left[\gamma\|\xi'\|\right]\right] + \frac{1}{n}\sum_{i=1}^n \mathbb{E}_R\left[\gamma\|\xi\|\right] \tag{168}$$

$$= 2\gamma\sqrt{d}\sigma, \tag{169}$$

we can write (166) as

$$\mathbb{E}_{\tilde{Z},R,S}\left(\mathcal{L}(f_\sigma, \tilde{Z}_S, R) - \mathcal{L}_{\mathrm{emp}}(f_\sigma, \tilde{Z}_S, R)\right)^2 \tag{170}$$

$$\ge \mathbb{E}_{\tilde{Z},S}\left(\mathcal{L}(f, \tilde{Z}_S) - \mathcal{L}_{\mathrm{emp}}(f, \tilde{Z}_S)\right)^2 - 4\gamma\sqrt{d}\sigma\,\mathbb{E}_{\tilde{Z},S}\left[\left|\mathcal{L}(f, \tilde{Z}_S) - \mathcal{L}_{\mathrm{emp}}(f, \tilde{Z}_S)\right|\right] \tag{171}$$

$$\ge \mathbb{E}_{\tilde{Z},S}\left(\mathcal{L}(f, \tilde{Z}_S) - \mathcal{L}_{\mathrm{emp}}(f, \tilde{Z}_S)\right)^2 - 4\gamma\sqrt{d}\sigma\sqrt{\mathbb{E}_{\tilde{Z},S}\left(\mathcal{L}(f, \tilde{Z}_S) - \mathcal{L}_{\mathrm{emp}}(f, \tilde{Z}_S)\right)^2}, \tag{172}$$

where the last line follows from Jensen's inequality ($(\mathbb{E}\,|X|)^2 \le \mathbb{E}\,X^2$). Summarizing, (158) and (172) relate expected generalization gap and expected squared generalization gap of $f_\sigma$ to those of $f$.

**Bounding expected generalization gap of $f$.**

$$\left|\mathbb{E}_{\tilde{Z},R,S}\left[\mathcal{L}(f_\sigma, \tilde{Z}_S, R) - \mathcal{L}_{\mathrm{emp}}(f_\sigma, \tilde{Z}_S, R)\right]\right| \tag{173}$$

$$\le \left|\mathbb{E}_{\tilde{Z},R,S}\left[\mathcal{L}(f_\sigma, \tilde{Z}_S, R) - \mathcal{L}_{\mathrm{emp}}(f_\sigma, \tilde{Z}_S, R)\right]\right| + 2\sqrt{d}\gamma\sigma \quad \text{(by (158))} \tag{174}$$

$$\le \frac{1}{n}\sum_{i=1}^n \mathbb{E}_{\tilde{z}\sim\tilde{Z}}\sqrt{2I(f_\sigma(\tilde{z}_S, \tilde{x}_i, R); S_i \mid S_{-i})} + 2\sqrt{d}\gamma\sigma \quad \text{(by Thm. 3.2)} \tag{175}$$

$$\le \frac{1}{n}\sum_{i=1}^n \mathbb{E}_{\tilde{z}\sim\tilde{Z}}\sqrt{\begin{array}{l}\frac{1}{2}\mathrm{KL}\left(f_\sigma(\tilde{z}_{S^{i\leftarrow 1}}, \tilde{x}_i, R)|S_{-i} \parallel f_\sigma(\tilde{z}_{S^{i\leftarrow 0}}, \tilde{x}_i, R)|S_{-i}\right) \\ +\frac{1}{2}\mathrm{KL}\left(f_\sigma(\tilde{z}_{S^{i\leftarrow 0}}, \tilde{x}_i, R)|S_{-i} \parallel f_\sigma(\tilde{z}_{S^{i\leftarrow 1}}, \tilde{x}_i, R)|S_{-i}\right)\end{array}} + 2\sqrt{d}\gamma\sigma \tag{176}$$

$$= \frac{1}{n}\sum_{i=1}^n \mathbb{E}_{\tilde{z}\sim\tilde{Z}}\sqrt{\frac{1}{2\sigma^2}\mathbb{E}_{S_{-i}}\|f(\tilde{z}_{S^{i\leftarrow 0}}, \tilde{x}_i) - f(\tilde{z}_{S^{i\leftarrow 1}}, \tilde{x}_i)\|_2^2} + 2\sqrt{d}\gamma\sigma \tag{177}$$

$$\le \frac{1}{n}\sum_{i=1}^n \sqrt{\frac{1}{2\sigma^2}\mathbb{E}_{\tilde{Z},S_{-i}}\left\|f(\tilde{Z}_{S^{i\leftarrow 0}}, \tilde{X}_i) - f(\tilde{Z}_{S^{i\leftarrow 1}}, \tilde{X}_i)\right\|_2^2} + 2\sqrt{d}\gamma\sigma \tag{178}$$

$$\le \sqrt{\frac{\beta^2}{\sigma^2}} + 2\sqrt{d}\gamma\sigma \quad \text{(by $\beta$ self-stability of $f$).} \tag{179}$$

Picking $\sigma^2 = \frac{\beta}{2\sqrt{d}\gamma}$, we get

$$\left|\mathbb{E}_{\tilde{Z},R,S}\left[\mathcal{L}(f_\sigma, \tilde{Z}_S, R) - \mathcal{L}_{\mathrm{emp}}(f_\sigma, \tilde{Z}_S, R)\right]\right| \le 2^{\frac{3}{2}}d^{\frac{1}{4}}\sqrt{\gamma\beta}. \tag{180}$$

**Bounding expected squared generalization gap of $f$.** To shorten the writing, let us denote $\mathbb{E}_{\tilde{Z},S}\left(\mathcal{L}(f,\tilde{Z}_S) - \mathcal{L}_{\text{emp}}(f,\tilde{Z}_S)\right)^2$ with $G$. Starting with (172), we get

$$G \leq \mathbb{E}_{\tilde{Z},R,S}\left(\mathcal{L}(f_\sigma,\tilde{Z}_S,R) - \mathcal{L}_{\text{emp}}(f_\sigma,\tilde{Z}_S,R)\right)^2 + 4\gamma\sqrt{d}\sigma\sqrt{G} \tag{181}$$

$$\leq \frac{8}{n}\left(\mathbb{E}_{\tilde{z}\sim\tilde{Z}}\left[\sum_{i=1}^n I(f_\sigma(\tilde{z}_S,\tilde{x},R);S_i \mid S_{-i})\right] + 2\right) + 4\gamma\sqrt{d}\sigma\sqrt{G} \quad \text{(by Thm. 3.2)} \tag{182}$$

$$\leq \frac{16}{n} + \frac{8}{n}\sum_{i=1}^n\left(\begin{array}{c}\frac{1}{4}\text{KL}\left(f_\sigma(\tilde{z}_{S^{i\leftarrow 1}},\tilde{x},R)|S_{-i} \parallel f_\sigma(\tilde{z}_{S^{i\leftarrow 0}},\tilde{x},R)|S_{-i}\right) \\ +\frac{1}{4}\text{KL}\left(f_\sigma(\tilde{z}_{S^{i\leftarrow 0}},\tilde{x},R)|S_{-i} \parallel f_\sigma(\tilde{z}_{S^{i\leftarrow 1}},\tilde{x},R)|S_{-i}\right)\end{array}\right) + 4\gamma\sqrt{d}\sigma\sqrt{G} \tag{183}$$

$$= \frac{16}{n} + \frac{8}{n}\sum_{i=1}^n E_{\tilde{Z},S}\left[\frac{1}{4\sigma^2}\left\|f(\tilde{Z}_{S^{i\leftarrow 0}},\tilde{X}) - f(\tilde{Z}_{S^{i\leftarrow 1}},\tilde{X})\right\|_2^2\right] + 4\gamma\sqrt{d}\sigma\sqrt{G} \tag{184}$$

$$\leq \frac{16}{n} + \frac{2}{\sigma^2}\left(2\beta^2 + n\beta_1^2 + n\beta_2^2\right) + 4\gamma\sqrt{d}\sigma\sqrt{G}. \tag{185}$$

The optimal $\sigma$ is given by

$$\sigma = \left(\frac{2\beta^2 + n\beta_1^2 + n\beta_2^2}{\gamma\sqrt{G}\sqrt{d}}\right)^{\frac{1}{3}}, \tag{186}$$

and gives

$$G \leq \frac{16}{n} + 6d^{\frac{1}{3}}\gamma^{\frac{2}{3}}\left(2\beta^2 + n\beta_1^2 + n\beta_2^2\right)^{\frac{1}{3}} G^{\frac{1}{3}}. \tag{187}$$

We discuss 2 cases.

(i) $\frac{16}{n} \geq 6d^{\frac{1}{3}}\gamma^{\frac{2}{3}}\left(2\beta^2 + n\beta_1^2 + n\beta_2^2\right)^{\frac{1}{3}} G^{\frac{1}{3}}$. In this case $G \leq \frac{32}{n}$.

(ii) $\frac{16}{n} < 6d^{\frac{1}{3}}\gamma^{\frac{2}{3}}\left(2\beta^2 + n\beta_1^2 + n\beta_2^2\right)^{\frac{1}{3}} G^{\frac{1}{3}}$. In this case, we have

$$G \leq 12d^{\frac{1}{3}}\gamma^{\frac{2}{3}}\left(2\beta^2 + n\beta_1^2 + n\beta_2^2\right)^{\frac{1}{3}} G^{\frac{1}{3}}, \tag{188}$$

which simplifies to

$$G \leq 12^{\frac{3}{2}}\sqrt{d}\gamma\sqrt{2\beta^2 + n\beta_1^2 + n\beta_2^2}. \tag{189}$$

Combining these cases we can write that

$$G \leq \max\left\{\frac{32}{n}, 12^{\frac{3}{2}}\sqrt{d}\gamma\sqrt{2\beta^2 + n\beta_1^2 + n\beta_2^2}\right\} \tag{190}$$

$$\leq \frac{32}{n} + 12^{\frac{3}{2}}\sqrt{d}\gamma\sqrt{2\beta^2 + n\beta_1^2 + n\beta_2^2}. \tag{191}$$

**Remark 1.** The bounds of this theorem work even when $\mathcal{Y} = \mathcal{K} = [a,b]^d$ instead of $\mathbb{R}^d$. To see this, we first clip the noisy predictions to be in $[a,b]^d$:

$$f_\sigma^c(z,x)_i \triangleq \text{clip}(f_\sigma(z,x),a,b)_i, \quad \forall i \in [d]. \tag{192}$$

Inequalities (158) and (172) that relate the expected generalization gap and expected squared generalization gap of $f_\sigma$ to those of $f$ stay true when replacing $f_\sigma$ with $f_\sigma^c$. Furthermore, by data processing inequality, mutual informations measured with $f_\sigma^c$ can always be upper bounded by the corresponding mutual informations informations measures with $f_\sigma$. Therefore, generalization bounds that hold for $f_\sigma$ will also for $f_\sigma^c$, allowing us to follow the exact same proofs above.

**Remark 2.** In the construction of $f_\sigma$ we used Gaussian noise with zero mean and $\sigma^2 I$ covariance matrix. A natural question arises whether a different type of noise would give better bounds. Inequalities (158) and (172) only use the facts that noise components are independent, have zero-mean and $\sigma^2$ variance. Therefore, if we restrict ourselves to noise distributions with independent components, each

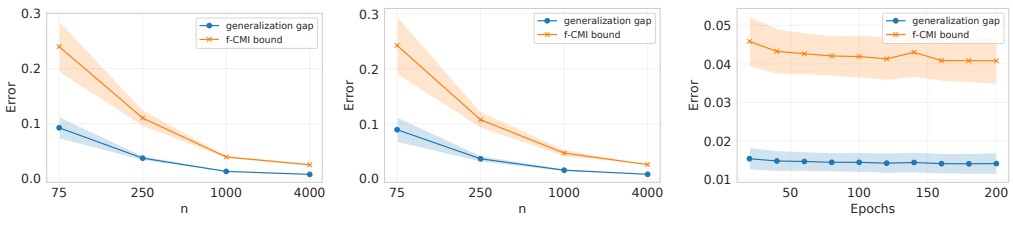

| | | |
|---|---|---|
| (a) using 4 times wider network | (b) training with a random seed | (c) dynamics of the bound |

Figure 3: Comparison of expected generalization gap and $f$-CMI bound for MNIST 4 vs 9 classification with a 4-layer CNN. The figure on the left repeats the experiment of Fig. 1a (i.e., MNIST 4 vs 9 classification with the CNN described in Table 1 trained using a deterministic training algorithm) while modifying the network to have 4 times more neurons at each layer. The figure in the middle repeats the experiment of Fig. 1a while making the training algorithm stochastic by randomizing the seed. The figure on the right corresponds to the experiment of Fig. 1a and plots the expected generalization gap and the f-CMI bound versus the training time.

| Layer type | Parameters |
|---|---|
| Conv | 32 filters, $4 \times 4$ kernels, stride 2, padding 1, batch normalization, ReLU |
| Conv | 32 filters, $4 \times 4$ kernels, stride 2, padding 1, batch normalization, ReLU |
| Conv | 64 filters, $3 \times 3$ kernels, stride 2, padding 0, batch normalization, ReLU |
| Conv | 256 filters, $3 \times 3$ kernels, stride 1, padding 0, batch normalization, ReLU |
| FC | 128 units, ReLU |
| FC | 2 units, linear activation |

Table 1: The architecture of the 4-layer convolutional neural network used in MNIST 4 vs 9 classification tasks. The wider version of this network is constructed by multiplying the number of channels/neurons of each hidden layer by 4.

of which has zero mean and $\sigma^2$ variance, then the best bounds will be produced by noise distributions that result in the smallest KL divergence of form $\mathrm{KL}\left(f_\sigma(\tilde{z}_{S^{i \leftarrow 1}}, x, R)|S_{-i} \parallel f_\sigma(\tilde{z}_{S^{i \leftarrow 0}}, x, R)|S_{-i}\right)$. An informal argument below hints that the Gaussian distribution might be the optimal choice of the noise distribution when $f_\sigma(\tilde{z}_{S^{i \leftarrow 1}}, x, R)$ and $f_\sigma(\tilde{z}_{S^{i \leftarrow 0}}, x, R)$ are close to each other.

Let us fix $\sigma^2$ and consider two means $\mu_1 < \mu_2 \in \mathbb{R}$. Let $\mathcal{F} = \{p(x; \mu) \mid \mu \in \mathbb{R}\}$ be a family of probability distributions with one mean parameter $\mu$, such that every distribution of it has variance $\sigma^2$ and KL divergences between members of $\mathcal{F}$ exist. Let $X_1 \sim p(x, \mu_1)$ and $X_2 \sim p(x, \mu_2)$. We are interested in finding such a family $\mathcal{F}$ that $\mathrm{KL}(X \parallel Y)$ is minimized. For small $\mu_2 - \mu_1$, we know that

$$\mathrm{KL}(X \parallel Y) \approx \frac{1}{2}(\mu_2 - \mu_1)\mathcal{I}(\mu_1)(\mu_2 - \mu_1), \tag{193}$$

where $\mathcal{I}(\mu)$ is the Fisher information of $p(x; \mu)$. Furthermore, let $\widehat{\mu_1} \triangleq X$. As $\mathbb{E}\,\widehat{\mu_1} = \mu_1$ and $\mathrm{Var}[\widehat{\mu_1}] = \sigma^2$, the Cramer-Rao bound gives

$$\sigma^2 = \mathrm{Var}[\widehat{\mu_1}] \geq \frac{1}{\mathcal{I}(\mu_1)}. \tag{194}$$

This gives us the following lower bound on the KL divergence between $X$ and $Y$:

$$\mathrm{KL}(X \parallel Y) \gtrapprox \frac{1}{2\sigma^2}(\mu_2 - \mu_1)^2, \tag{195}$$

which is matched by the Gaussian distribution.

# B  Experiment details and additional results

In this appendix we present additional experimental results and details that were not included in the main text due to space constraints.

**Estimation of generalization gap.** In all experiments we draw $k_1$ samples of $\tilde{Z}$, each time by randomly drawing $2n$ examples from the corresponding dataset and grouping then into $n$ pairs. For each sample $\tilde{z}$, we draw $k_2$ samples of the training/test split variable $S$ and randomness $R$. We then run the training algorithm on these $k_2$ splits (in total $k_1 k_2$ runs). For each $\tilde{z}$, $s$ and $r$, we estimate the population risk with the average error on the test examples $\tilde{z}_{\text{neg}(s)}$ and get an estimate of the generalization gap $\mathcal{L}(f, \tilde{z}_s, r) - \mathcal{L}_{\text{emp}}(f, \tilde{z}_s, r)$. For each $\tilde{z}$, we average over the $k_2$ samples of $S$ and $R$ to get an estimate $\hat{g}(\tilde{z})$ of $\mathbb{E}_{S,R}[\mathcal{L}(f, \tilde{z}_s, R) - \mathcal{L}_{\text{emp}}(f, \tilde{z}_s, R)]$. Note that this latter quantity is not the expected generalization gap yet, as it still misses an expectation over $\tilde{z}$. Figures 1 and 3 plot the mean and the standard deviation of $\hat{g}(\tilde{z})$, estimated using the $k_1$ samples of $\tilde{Z}$. Note that this mean will be an unbiased estimate of the true expected generalization gap $\mathbb{E}_{\tilde{Z},S,R}[\mathcal{L}(f, \tilde{z}_s, r) - \mathcal{L}_{\text{emp}}(f, \tilde{z}_s, r)]$.

**Estimation of $f$-CMI bound.** Similarly, for each $\tilde{z}$ we use the $k_2$ samples of $S$ and $R$ to estimate $f\text{-CMI}(f, \tilde{z}, \{i\}) = I(f(\tilde{z}_S, \tilde{x}_i, R); S_i)$, $i \in [n]$. As in all considered cases we deal with classification problems (i.e., having discrete output variables), this is done straightforwardly by estimating all the states of the joint distribution of $f(\tilde{z}_S, \tilde{x}_i, R)$ and $S_i$, and then using a plug-in estimator of mutual information. The bias of this plug-in estimator is $O\left(\frac{1}{k_2}\right)$, while the variance is $O\left(\frac{(\log k_2)^2}{k_2}\right)$ [24]. To estimate $f\text{-CMI}_{\mathcal{D}}(f, \{i\}) = \mathbb{E}_{\tilde{z} \sim \tilde{Z}}[f\text{-CMI}(f, \tilde{z}, \{i\})]$ we use $k_1$ samples of $\tilde{Z}$. After this step the estimation bias stays the same, while the variance increases by $O\left(\frac{1}{k_1}\right)$. The error bars in Figures 1 and 3 are empirical standard deviations computed using these $k_1$ estimates.

In the main text we consider the three experiments: (a) MNIST 4 vs 9 classification using standard training algorithms, (b) MNIST 4 vs 9 classification using SGLD, and (c) CIFAR-10 classification with fine-tuned pretrained ResNet-50. The experimental details of these experiments are presented in Tables 2, 3, and 4, respectively. In all cases the loss function was the cross-entropy loss. We did the experiments on a cluster of 4 computers, each with 4 NVIDIA GeForce RTX 2080 Ti GPUs. Training only single-GPU models, the experiments take 2-3 days.

| | |
|---|---|
| Network | The 4-layer CNN described in Table 1. |
| Optimizer | ADAM with $0.001$ learning rate and $\beta_1 = 0.9$. |
| Batch size | 128 |
| Number of examples ($n$) | [75, 250, 1000, 4000] |
| Number of epochs | 200 |
| Number of samples for $\tilde{Z}$ ($k_1$) | 5 |
| Number of samplings for $S$ for each $\tilde{z}$ ($k_2$) | 30 |

Table 2: Experimental details for MNIST 4 vs 9 classification in case of the standard training algorithm.

| | |
|---|---|
| Network | The 4-layer CNN described in Table 1. |
| Learning rate schedule | Starts at $0.004$ and decays by a factor of $0.9$ after each 100 iterations. |
| Inverse temperature schedule | $\min(4000, \max(100, 10e^{t/100}))$, where $t$ is the iteration. |
| Batch size | 100 |
| Number of examples ($n$) | 4000 |
| Number of epochs | 40 |
| Number of samples for $\tilde{Z}$ ($k_1$) | 5 |
| Number of samplings for $S$ for each $\tilde{z}$ ($k_2$) | 30 |

Table 3: Experimental details for MNIST 4 vs 9 classification in case of the SGLD training algorithm.

| | |
|---|---|
| Network | ResNet-50 pretrained on ImageNet. |
| Optimizer | SGD with $0.01$ learning rate and $0.9$ momentum. |
| Data augmentations | Random horizontal flip and random 28x28 cropping. |
| Batch size | 64 |
| Number of examples ($n$) | [1000, 5000, 20000] |
| Number of epochs | 40 |
| Number of samples for $\tilde{Z}$ ($k_1$) | 1 |
| Number of samplings for $S$ for each $\tilde{z}$ ($k_2$) | 40 |

Table 4: Experimental details for CIFAR-10 classification using fine-tuned ResNet-50 networks.

| | |
|---|---|
| Network | ResNet-50 pretrained on ImageNet. |
| Data augmentations | Random horizontal flip and random 28x28 cropping. |
| Learning rate schedule | Starts at 0.01 and decays by a factor of 0.9 after each 300 iterations. |
| Inverse temperature schedule | $\min(16000, \max(100, 10e^{t/300}))$, where $t$ is the iteration. |
| Batch size | 64 |
| Number of examples ($n$) | 20000 |
| Number of epochs | 16 |
| Number of samples for $\tilde{Z}$ ($k_1$) | 1 |
| Number of samplings for $S$ for each $\tilde{z}$ ($k_2$) | 40 |

Table 5: Experimental details for CIFAR-10 classification experiment, where a pretrained ResNet-50 is fine-tuned using the SGLD algorithm.