# OpenReview forum: "Information-theoretic generalization bounds for black-box learning algorithms"
_NeurIPS.cc/2021/Conference — NeurIPS 2021 Poster_

### Official Review · Reviewer_FhFb · 2021-07-11

**Rating:** 5
**Confidence:** 4

**Summary:**

This paper proposes new information-theoretic generalization bounds. These bounds extend some existing results and seem easier to compute. The authors also demonstrate their bounds through some applications and experiments.

**Limitations And Societal Impact:**

See my above comments.

**Main Review:**

Strengths:
Using the information-theoretic framework, introduced by Xu and Raginsky, for studying the generalization error is interesting.

The new generalization bounds in Section 3 seem easier to compute compared with existing mutual-information-based generalization bounds.

The experimental results seem to suggest that the new bounds in this paper are promising.

Weaknesses:

My main concern of this paper is its novelty and unclear comparison with existing literature.

1. The comparison with existing literature is unclear. For example, in the abstract, the authors claimed that their bounds improve over the existing information-theoretic bounds since (a) they give meaningful results for deterministic algorithms and (b) they are significantly easier to estimate. In the second paragraph of the paper, the authors echoed their claim and provided some references. However, it seems that many existing papers have already contributed to the above two challenges. In fact, the generalization bounds in [4,28] are applicable to deterministic algorithms, and the bounds in [11,19,20] seem easy to compute as well. There are some missing references that also contribute to these two directions. Hence, I recommend the author revise their introduction and clarify their contributions.

2. The authors provided a generalization bound in Theorem 2.2 and stated that this bound matches the result of Bu et al. [4] when m=1. Then authors proved that this bound is non-decreasing in m. Hence, if I understand correctly, Bu et al. have already derived the tightest case. Then what is the purpose of introducing other weaker bounds? In the same vein, bounds with similar forms have already appeared before. For example, Raginsky has presented a similar result in a tutorial at NASIT 2019 (see the recorded video at https://www.itsoc.org/conferences/schools/nasit2019 1:25:05). Negrea et al introduced a similar bound. Although their bound is slightly weaker than Thm 2.2. in this paper, it does not seem that the improvement is significant.

Similar comments also apply to some bounds in Section 2.2.

3. Section 3. The authors considered a different framework where they assumed that the learning method implements a function f: Z^n \times X \times R \to K. This framework has been considered in the following missing reference for studying excess risk. Hence, it seems that the authors just applied the existing bounds introduced in [28] to this new framework.

Xu, A. and Raginsky, M., 2020. Minimum excess risk in Bayesian learning. arXiv preprint arXiv:2012.14868.

My second concern is the experiments in this paper.
1. The authors compared their bound with the one in Negrea et al [19] in Fig 1 (b). For the first 12 epochs, the bound in Negrea et al [19] outperforms the bound in this paper. Given that the SGLD algorithm converges in 16 epochs, it seems that the new bound introduced in this paper is often weaker than the bound in Negrea et al [19] before the algorithm converges.

2. The bound in Negrea et al [19] blows up after the algorithm convergences and that’s the region where the bound in this paper is sharper. However, a follow-up work [11] has already proposed a new generalization bound which seems to address this issue and becomes much tighter. This bound is extended to the setting of SGLD in the following missing reference. Hence, I wonder how the generalization bound in this paper compares with these new bounds.

Rodríguez-Gálvez, B., Bassi, G., Thobaben, R. and Skoglund, M., 2021, April. On random subset generalization error bounds and the stochastic gradient Langevin dynamics algorithm. In 2020 IEEE Information Theory Workshop (ITW) (pp. 1-5). IEEE.

3. The generalization bound in this paper is decreasing w.r.t. the number of epochs in Fig. 1. However, it seems that the true generalization gap is increasing which is captured by the bound in Negrea et al [19]. How can you interpret this phenomenon?

Other comments

The generalization bounds in Eq.(29). It is unclear to me why the second term has a $n$ inside the square root. How to explain this dependence? Does that suggest that the generalization gap is increasing w.r.t. the sample size. Also, how does this bound compare with other existing stability bounds?

The authors provided some high-probability bounds (e.g., Eq. 3 and Line 88) in the paper. I wonder how they compare with the one in [34]. It seems that using the “monitor technique” can lead to a tighter bound compared with the bound obtained from Markov’s inequality. Also, how do these high-probability bounds compare with the ones in the following reference?
Esposito, A.R., Gastpar, M. and Issa, I., 2021. Generalization error bounds via Rényi-, f-Divergences and Maximal Leakage. IEEE Transactions on Information Theory.

Line 112—113. The authors wrote “While these bounds are worse than the corresponding bounds of Thm. 2.2, it is sometimes easier to manipulate them analytically.” This is unclear to me. Can the authors elaborate more?

**Time Spent Reviewing:**

10

---

> ### Author Response · Authors · 2021-08-09
> **Response to Reviewer FhFb (Part 1)**
>
> We apologize for submitting the reply of this review later compared to the replies of the other 4 reviews. The reason for this delay was the additional experiments we conducted to supplement to this reply (see below).
>
> ### Response to the main concern: novelty and unclear comparison with existing literature.
> > The comparison with existing literature is unclear. For example, in the abstract, the authors claimed that their bounds improve over the existing information-theoretic bounds since (a) they give meaningful results for deterministic algorithms and (b) they are significantly easier to estimate. In the second paragraph of the paper, the authors echoed their claim and provided some references. However, it seems that many existing papers have already contributed to the above two challenges. In fact, the generalization bounds in [4,28] are applicable to deterministic algorithms, and the bounds in [11,19,20] seem easy to compute as well.
>
> We agree that many good contributions have been done towards the mentioned two challenges. However, we do think that most of the existing bounds are either hard to estimate or may be vacuous for deterministic algorithms. For example, the bounds of Xu and Raginsky [34], Negrea et el. [19], and Bu et al. [4] are hard to estimate as they involve estimation of mutual information between a high-dimensional non-discrete variable $W$ and at least one example $Z_i$. Furthermore, this mutual information can be infinite in case of deterministic algorithms or when $H(Z_i)$ is infinite.
> The bounds of Haghifam et al. [11] and Steinke and Zakynthinou [28] are also hard to estimate for general algorithms, as they involve estimation of mutual information between $W$ and at least one train-test split variable $S_i$.
> The bound of Neu [20] is relatively easier to compute but is applicable only to SGD.
> Similarly, Negrea et al. [19] and Haghifam et al. [11] provide bounds that are relatively easier to compute, but are applicable only to SGLD.
> The bound of Steinke and Zakynthinou [28] is applicable to deterministic algorithms but will give a trivial bound (a constant bound) in all cases when the bound of Bu et al. [4] is equal to infinity.
> In summary, all of these mentioned bounds are either (a) designed for one algorithm, (b) are hard to estimate, or (c) are not applicable to general deterministic algorithms.
> In contrast, our main result of Corollary 2 holds for all algorithms, including deterministic ones, and is easy to estimate if the predictions are low-dimensional variables.
>
>
> > There are some missing references that also contribute to these two directions.
>
> Please let us know which papers you think are relevant and were not cited. We are happy to discuss them in future revisions.
>
>
> > The authors provided a generalization bound in Theorem 2.2 and stated that this bound matches the result of Bu et al. [4] when m=1. Then authors proved that this bound is non-decreasing in m. Hence, if I understand correctly, Bu et al. have already derived the tightest case. Then what is the purpose of introducing other weaker bounds? In the same vein, bounds with similar forms have already appeared before. For example, Raginsky has presented a similar result in a tutorial at NASIT 2019 (see the recorded video at https://www.itsoc.org/conferences/schools/nasit2019 1:25:05). Negrea et al introduced a similar bound. Although their bound is slightly weaker than Thm 2.2. in this paper, it does not seem that the improvement is significant.
>
> In section 2 we describe existing weigh-based information-theoretic generalization bounds, slightly generalize or improve some of them, and prove some relations between them. We do not consider the results of this section as our main contribution. Additionally, in section 2 we provide bounds of expected squared generalization gap in multiple previously considered settings and stress the non-trivial differences in bounding expected generalization gap and bounding expected squared generalization gap. In the case of the latter, one cannot easily derive bounds where information is measured with subsets or singletons of $Z_i$s of $S_i$s.
>  Overall, the purpose of this section is to introduce existing bounds, show that they can be derived in unified fashion, and prepare grounds for the f-CMI bounds introduced in section 3, which we consider our main contribution. We will clarify our contributions and explain the structure of the paper better in future revisions.
>
>
> > Section 3. The authors considered a different framework where they assumed that the learning method implements a function f: Z^n \times X \times R \to K. This framework has been considered in the following missing reference for studying excess risk. Hence, it seems that the authors just applied the existing bounds introduced in [28] to this new framework.
>
> We are well aware of the mentioned work. Indeed, in that work too, the learning method implements a function $f: \mathcal{Z}^n \times \mathcal{X} \times \mathbb{R} \to \mathcal{K}$. However, the scope of that paper is completely different from ours. Our main results cannot be seen as just a simple extension of bounds of Steinke and Zakynthinou [28] to this formulation of learning methods. The key improvements over [28] are the main f-CMI bound of Corollary 2, Theorem 3.2 (functional stability-based bounds), Theorem 4.1 (a stronger result in case of learning with finite VC-dimension hypothesis classes), and Theorem 4.2 (applications to stable algorithms).
>
> Furthermore, in the course of doing this work, we arrived to the setting of Steinke and Zakynthinou [28] naturally and independently, starting from the idea of moving from the weight-space perspective to a function-space perspective in the bound of Xu and Raginsky [34] (please see the last paragraph of page 3).
> The setting of Steinke and Zakynthinou [28] was originally introduced for a different motivation -- normalizing sample information of each example to be 1 bit.

---

> ### Author Response · Authors · 2021-08-09
> **Response to Reviewer FhFb (Part 2)**
>
> ### Response to the second concern: experiments
>
> > a) The authors compared their bound with the one in Negrea et al [19] in Fig 1 (b). For the first 12 epochs, the bound in Negrea et al [19] outperforms the bound in this paper. Given that the SGLD algorithm converges in 16 epochs, it seems that the new bound introduced in this paper is often weaker than the bound in Negrea et al [19] before the algorithm converges.
> > b) The bound in Negrea et al [19] blows up after the algorithm convergences and that’s the region where the bound in this paper is sharper. However, a follow-up work [11] has already proposed a new generalization bound which seems to address this issue and becomes much tighter. This bound is extended to the setting of SGLD in the following missing reference. Hence, I wonder how the generalization bound in this paper compares with these new bounds.
>
> Indeed, in the early stages of training the proposed bound is larger than the bound of Negrea et al [19], which correctly predicts that generalization gap is small, as it is designed specifically for SGLD and "knows" that network weights cannot travel too far from their initialization point in short training time.
> However, in practice training will not be stopped in such early training stages. In our particular case shown in Figure 1, most early stopping methods will stop after epoch 12, starting where the proposed bound outperforms the bound of Negrea et al. [19].
>
> As the considered classification task in Figure 1b is too simple (MNIST 4 vs 9 classification with a CNN), we tried fine-tuning a pretrained ResNet-50 with SGLD on CIFAR-10 with $n=20000$ examples (as in Figure 1c). In this new experiment we used similar learning rate and inverse temperature schedules as in the MNIST experiment. The results are presented in the table below. Note that even at epoch 2, the f-CMI bound is already much better than the Negrea et al. bound.
>
> | | | | | | | | | |
> |:--|--|--|--|--|--|--|--|--|
> | Epoch | 2 | 4 | 6 | 8 | 10 | 12 | 14 | 16 |
> | Test error | 0.060 | 0.048 | 0.044 | 0.042 | 0.041 | 0.040 | 0.040 | 0.039 |
> | Generalization gap | 0.022 | 0.032 | 0.036 | 0.038 | 0.038 | 0.039 | 0.039 | 0.039 |
> | Negrea et al. bound | 0.456 | 0.653 | 1.160 | 2.101 | 2.624 | 2.877 | 3.023 | 3.122 |
> | f-CMI bound | 0.101 | 0.100 | 0.103 | 0.105 | 0.105 | 0.105 | 0.106 | 0.106 |
>
> We did not compare the f-CMI bound with the bound of Haghifam et al. [11] for the following reasons: (a) the experiments in their paper show that the difference between these two bounds is not large; (b) the bound of Haghifam et al. [11] is qualitatively similar to that of Negrea et al. [19] (for example, both depend on gradient variance and increase with T); and (c) we found it harder to implement. We will try to add comparison with the bound of Haghifam et al.
>
> > The generalization bound in this paper is decreasing w.r.t. the number of epochs in Fig. 1. However, it seems that the true generalization gap is increasing which is captured by the bound in Negrea et al [19]. How can you interpret this phenomenon?
>
> One possible explanation is there are many examples that are learned significantly earlier when they are included in the training set, compared to the case when they are not included. At early stages of learning, the prediction confidences on such examples will be correlated with their membership information (i.e. whether an example is in the training or test set). Small prediction confidence differences on these examples may have small effects on the generalization gap, but large effects on the f-CMI bound, as these differences might reveal significant amount of membership information. This indicates an area of improvement for the proposed bounds.
>
> ### Response to other comments
> > The generalization bounds in Eq.(29). It is unclear to me why the second term has a $n$ inside the square root. How to explain this dependence? Does that suggest that the generalization gap is increasing w.r.t. the sample size. Also, how does this bound compare with other existing stability bounds?
>
> The dependence of $\beta$, $\beta_1$ and $\beta_2$ on $n$ is implicit. All of them are expected to decrease as we increase $n$. The bound of equation (29) will converge to zero as $n \rightarrow \infty$ if $\beta$ is $o(1)$, while $\beta_1$ and $\beta_2$ are $o(\frac{1}{\sqrt{n}})$.
>
> (Copying from the response to Reviewer AGS2.) The notion of pointwise hypothesis stability $\beta_2$ defined by Bousquet and Elisseeff [3] (definition 4) is comparable to our notion of self-stability $\beta$.
> The first part of Theorem 11 in [3] describes a generalization bound, where the difference between empirical and population losses is of order $\sqrt{\beta_2}$, which is comparable with our result of Thm. 4.2 ($\Theta(\sqrt{\beta})$). The proof there also contains a bound on the expected squared difference of empirical and population losses. That bound is of order $\frac{1}{n} + \beta_2$. In contrast, our result in eq. (29) contains two extra terms related test-stability and train-stability (the terms $n \beta_1^2$ and $n \beta_2^2$). It is expected that $\beta_1$ and $\beta_2$ defined in our paper are much smaller than $\beta$. If $\beta$ dominates $n \beta_1^2 + b \beta_2^2$, then the bound of eq. (29) will match the classical result of [3].
> On the other hand, the results derived with uniform stability in [3], which is a much stronger condition, are quadratically better and provide high-probability bounds. Whether such results can be derived from the mutual-information-based bounds is unclear.
> We will research more on this topic, and add some discussion on the comparison of the results of Thm. 4.2 with the literature.
>
> Furthermore, as discussed in the paper, it is still an open question whether one can remove the terms $n \beta_1^2$ and $n \beta_2^2$ from Eq. (29). These terms arose because the technique of moving from measuring information with $S$ to measuring information with $S_i$ is not applicable for bounding expected squared generalization gap (in contrast to the case of bounding expected generalization gap).
>
> > The authors provided some high-probability bounds (e.g., Eq. 3 and Line 88) in the paper. I wonder how they compare with the one in [34]. It seems that using the “monitor technique” can lead to a tighter bound compared with the bound obtained from Markov’s inequality. Also, how do these high-probability bounds compare with the ones in the following reference? Esposito, A.R., Gastpar, M. and Issa, I., 2021. Generalization error bounds via Rényi-, f-Divergences and Maximal Leakage. IEEE Transactions on Information Theory.
>
> The tail bound of Eq. (3) will produce a result comparable with the result of Theorem 3 of Xu and Raginsky [34] when applied in the setting of Theorem 2.1 of our work. In the notation of [34], the result we will get that corresponds to the result of Theorem 3 of [34] will be $n = \frac{4\sigma^2}{\alpha^2}\left(\frac{\epsilon}{\beta} + \frac{\log 3}{\beta}\right)$. Note that the difference is in the second term inside the parentheses. Instead of $\frac{\log 3}{\beta}$, the result of [34] contains $\log\frac{2}{\beta}$. However, this improvement is not significant as the first term $\frac{\epsilon}{\beta}$ will dominate the second term whenever $\epsilon = \Omega(1)$ (i.e., capturing at least constant number of bits about the training dataset).
>
> Thanks for the reference [\*], we will include it the discussions. Judging by the rightmost column of Table 1 of [\*], the tail bounds (or alternatively the sample complexity bounds) achieved by using Shannon mutual information are worse than the results achieved using Maximal Leakage, $\alpha$-Sibson's MI, and $\chi^2$ divergence, as the latter ones have logarithmic dependence on $\delta$ (the error probability, denoted by $\beta$ in the previous paragraph).
>
>
> [*] Rodríguez-Gálvez, B., Bassi, G., Thobaben, R. and Skoglund, M., 2021, April. On random subset generalization error bounds and the stochastic gradient Langevin dynamics algorithm. In 2020 IEEE Information Theory Workshop (ITW) (pp. 1-5). IEEE.
>
> > Line 112—113. The authors wrote “While these bounds are worse than the corresponding bounds of Thm. 2.2, it is sometimes easier to manipulate them analytically.” This is unclear to me. Can the authors elaborate more?
>
> The intuition is that quantities like $I(W; Z_i)$ are hard to manipulate analytically as they involve marginalization over $Z_{-i}$. When conditioning on $Z_{-i}$, we are left with only one two sources of randomness: $Z_i$ and $R$. Therefore, the quantity $I(W; Z_i \mid Z_{-i})$ becomes easier to manipulate. Note that in the proof of Thm. 4.2 we start with standard f-CMI bounds, but then upper bound them with stability-based bounds of Thm. 3.2, as they are easier to manipulate and allow us to reach to Proposition 4.

---

> > ### Comment · Reviewer_FhFb · 2021-08-10
> > **Follow-up questions**
> >
> > Thanks for the thorough response to my comments! I am glad that most of my comments have been clarified/addressed in the response. Nonetheless, I still have a few follow-up questions regarding the experiments in this paper.
> >
> > 1. The authors explained that they did not compare with Haghifam et al. because
> > >(a) the experiments in their paper show that the difference between these two bounds is not large; (b) the bound of Haghifam et al. [11] is qualitatively similar to that of Negrea et al. [19] (for example, both depend on gradient variance and increase with T); and (c) we found it harder to implement.
> >
> > For (a), it seems that their Figure 1 does show that the difference is large.
> > For (b) and (c), if the bound of Haghifam et al. is qualitatively similar to that of Negrea et al., then what is the challenge of implementing it?  Can the authors elaborate more?
> >
> > 2. The new experimental result included in the response is interesting. However, the concern I raised in my initial review still emerges: the generalization bound proposed in this paper cannot reflect (or predict) the true generalization phenomenon (i.e., the generalization error is increasing but the bound tends to be stable). This seems to be a limitation of this work.

---

> > > ### Author Response · Authors · 2021-08-11
> > > **Response to the follow-up questions**
> > >
> > > Thank you for the quick follow-up.
> > >
> > > > For (a), it seems that their Figure 1 does show that the difference is large. For (b) and (c), if the bound of Haghifam et al. is qualitatively similar to that of Negrea et al., then what is the challenge of implementing it? Can the authors elaborate more?
> > >
> > > It seems that the large differences between the bound of Haghifam et al. and Negrea et al. arise at the late stages of training, where the latter starts to blow up. For example, in the case of CIFAR with CNN, the larger differences between the two bounds start to occur around iteration 450. The main text specifies that around iteration 600, the training error becomes vanishingly small.
> > >
> > > Computing the bound of Haghifam et al. requires access to the all $n$ pairs of examples during training, in contrast to the standard scenario, where we need access to only the $n$ training examples.
> > > This created a minor technical difficulty, as our codebase was designed for the standard scenario. Nevertheless, as noted earlier, we will implement computation of the bound of Haghifam et al. and add comparisons with it in future revisions.
> > >
> > >
> > > > The new experimental result included in the response is interesting. However, the concern I raised in my initial review still emerges: the generalization bound proposed in this paper cannot reflect (or predict) the true generalization phenomenon (i.e., the generalization error is increasing but the bound tends to be stable). This seems to be a limitation of this work.
> > >
> > > We agree that in some cases our bound does not correctly reflect changes in generalization error.
> > > Both Fig. 1b and the new experiment indicate that the proposed bound is less tight in early stages of training.
> > > While this can be considered a limitation, in practice, this limitation is often avoided as the training is rarely stopped at such early stages.
> > > As hypothesized in our reply above, this kind of behavior might be stemming from the fact that mutual information ignores the geometry of predictions -- tiny differences in predictions that have small effect on generalization error can reveal significant membership information.

---

### Official Review · Reviewer_AGS2 · 2021-07-13

**Rating:** 6
**Confidence:** 2

**Summary:**

The authors build upon recent literature on information theoretic bounds on generalization error and specifically the conditional mutual information bound by Steinke and Zakynthinou [28]. They propose an improved bound that is tigther than [28] and easier to compute in practice. It is achieved by replacing the weight-parameter vector in the mutual information expression of [28] with a prediction function evaluated on a subset of in-sample (training) data and a subset of out-of-sample data.  This enables the authors to give performance guarantees on deterministic training algorithms. Several toy examples and experimental results are presented to illustrate the advantages of the improved generalization bound.

**Limitations And Societal Impact:**

The authors have not listed societal impact. Limitations have been discussed.

**Main Review:**

The proposed bound involving mutual information between the sample index variables and prediction function involving the first n samples is interesting and has practical advantages. However the proof seems to follow prior works and builds upon Lemma 1. Can the authors explain what aspects are non-trivial in the derivation of the new bound?

Can the authors comment on how the result in Thm. 4.2 compares with other generalization results on deterministic algorithms n the literature?

In the experimental section I was wondering why the authors did not compare their bound with bounds other than Negrea et al.? There are other state of the art bounds (such as the work of Haghifam et al) that are better than Negrea et al. for SGLD.  Are there difficulties in comparing with these works?

**Time Spent Reviewing:**

2 hrs

---

> ### Author Response · Authors · 2021-08-06
> **Response to AGS2**
>
> > The proposed bound involving mutual information between the sample index variables and prediction function involving the first n samples is interesting and has practical advantages. However the proof seems to follow prior works and builds upon Lemma 1. Can the authors explain what aspects are non-trivial in the derivation of the new bound?
>
> Indeed, most results in sections 2 and 3 build upon Lemma 1. Similar lemmas were key in derivation of the main results in [4], [10], [11], [19], [22], [24], [28], and [34]. One difference is that our version of Lemma 1 is slightly more general, as it also provides an upper bound on expected squared difference too (eq. 2). This later allows us to easily derive bounds on the expected squared difference of training and test errors in multiple settings. While derivation is simple in most cases, it uncovers non-trivial differences in bounding expected generalization gap and bounding expected squared generalization gap. In the case of the latter, one cannot easily derive bounds where information is measured with subsets or singletons of $Z_i$s or $S_i$s. Hence, we see key differences in pairs of equations (5) and (6), (8) and (9), (13) and (14), (18) and (19), (23) and its subsequent equation, and in (28) and (29).
>
> The derivation of the proof of Thm. 4.2 is non-trivial and uncovers an interesting fact -- the most optimal generalization bounds in the particular scheme considered in the proof of Thm. 4.2 are derived using Gaussian noise (see Remark 2 in Appendix A).
>
> > Can the authors comment on how the result in Thm. 4.2 compares with other generalization results on deterministic algorithms n the literature?
>
> The notion of pointwise hypothesis stability $\beta_2$ defined by Bousquet and Elisseeff [3] (definition 4) is comparable to our notion of self-stability $\beta$.
> The first part of Theorem 11 in [3] describes a generalization bound, where the difference between empirical and population losses is of order $\sqrt{\beta_2}$, which is comparable with our result of Thm. 4.2 ($\Theta(\sqrt{\beta})$). The proof there also contains a bound on the expected squared difference of empirical and population losses. That bound is of order $\frac{1}{n} + \beta_2$. In contrast, our result in eq. (29) contains two extra terms related test-stability and train-stability (the terms $n \beta_1^2$ and $n \beta_2^2$). It is expected that $\beta_1$ and $\beta_2$ defined in our paper are much smaller than $\beta$. If $\beta$ dominates $n \beta_1^2 + b \beta_2^2$, then the bound of eq. (29) will match the classical result of [3].
> On the other hand, the results derived with uniform stability in [3], which is a much stronger condition, are quadratically better and provide high-probability bounds. Whether such results can be derived from the information-theoretic bounds is an open question.
> We will research more on this topic, and add some discussion on the comparison of the results of Thm. 4.2 with the literature.
>
> > In the experimental section I was wondering why the authors did not compare their bound with bounds other than Negrea et al.? There are other state of the art bounds (such as the work of Haghifam et al) that are better than Negrea et al. for SGLD. Are there difficulties in comparing with these works?
>
> For general algorithms most information-theoretic generalization bounds are hard to compute or are infinite. (Copying from the response to Reviewer LtsL.) The bounds of Xu and Raginsky [ref. 34 in the paper], Negrea et el. [ref. 19 in the paper], and Bu et al. [ref. 4 in the paper] are hard to estimate as they involve estimation of mutual information between a high-dimensional non-discrete variable $W$ and at least one example $Z_i$. Furthermore, this mutual information can be infinite in case of deterministic algorithms or when $H(Z_i)$ is infinite.
> The bounds of Haghifam et al. [ref. 11 in the paper] and  Steinke and Zakynthinou [ref. 28 in the paper] are also hard to estimate as they involve estimation of mutual information between $W$ and at least one train-test split variable $S_i$.
>
> In case of SGLD, we agree with the reviewer that the bound of Haghifam et al. is better. We did not consider it for a few reasons. First, the experiments in their paper show that the difference between these two bounds is not large. Second, the bound of Haghifam et al. is qualitatively similar to that of Negrea et al. (for example, both depend on gradient variance and increase with T). Third, we found it harder to implement. We will try to add comparison with the bound of Haghifam et al.

---

### Official Review · Reviewer_RQQn · 2021-07-21

**Rating:** 7
**Confidence:** 3

**Summary:**

The paper derives information-theoretic bounds for learning algorithms. The bounds are based on the predictions of the learned model, rather than the weights (output), which makes them tighter by the data-processing inequality and can yield non-vacuous bounds in cases where weight-based mutual information bounds are vacuous. The authors validate the analysis with a few experiments.

**Limitations And Societal Impact:**

Yes

**Main Review:**

The paper derives information-theoretic bounds for learning algorithms. The bounds are based on the predictions of the learned model, rather than the weights (output), which makes them tighter by the data-processing inequality and can yield non-vacuous bounds in cases where weight-based mutual information bounds are vacuous.  The new bounds build upon the earlier works of Xu & Raginsky (2017) and the more recent work of Steinke & Zakynthinou (2020) using the notion of the conditional mutual information (CMI).

First, the authors present a theorem that generalizes a bound of  Xu & Raginsky (2017) (Theorem 2.2 in the paper) in which the mutual information is measured between the weights W and the full dataset S. Theorem 2.2 uses a subset of S. However, I don't see how this theorem is different from Theorem 2.1 since taking a proper subset of a training set (i.e. for any fixed choice of u in the theorem), the distribution of S_u is the same as the distribution of a training sample of size m drawn iid. Would you please clarify why Eq 5 is not equivalent to Eq 4?

The authors then discuss the choice of the subset size m. They argue that having a smaller m yields a tighter bound. However, there is an early work that makes the same argument and is not discussed in the paper. In Alabdulmohsin (2015),  m=1 is used to derive non-vacuous bounds and it is shown that those bounds were tight (see for instance [1] and the subsequent works summarized in [2]). Note that the difference between using the weights and using the function itself (decision boundary) was discussed there as well using data-processing. Since this is an earlier work that makes the same argument, it should be discussed.

The main contribution of this work is Theorem 3.1, in which the authors introduce the notion of functional CMI. It is analogous to the notion of CMI introduced by Steinke & Zakynthinou (2020) but uses the predictions rather than the weights (output). One interesting difference is on the connections between mutual information and the VC dimension. Earlier works (e.g. Alabdulmohsin (2018) and Steinke & Zakynthinou (2020)) prove that an ERM *exists* in a VC class that has a small mutual information between the weights and the data. Using functional CMI, the authors show that all functions in a VC class have a finite f-CMI, which is a neat result. The difference here is that by looking at the predictions, they rule out cases in which the data is encoded in the weights without impacting the decision boundary significantly.

The authors then discuss some implications. For example, they look into the case of ensemble learning and show that the f-CMI of an ensemble can be bounded using the f-CMI of the base models. However, this argument is not really useful because it does not explain why an ensemble method tends to generalize better. The bound for the ensemble is larger than the bound of any base classifier.

Finally, the authors validate the analysis with a few experiments. The authors argue that the f-CMI bound "closely tracks" the generalization gap but this is misleading because the difference is one order of magnitude and the fact it is decreasing with increasing sample size does not make it "tracking" the generalization gap. After all, all useful generalization bounds decrease with increasing sample size. I suggest that the authors rephrase that claim (perhaps by only arguing that the bound is non-vacuous). Also, in Line 319, the statement is false because weight-based mutual information bounds need not be vacuous (e.g. when m=1 as discussed earlier). Yes, they can be vacuous but need not be.

One minor comment. In Lines 128 and 129, I think there is a typo. Shouldn't "n" be "2n"?

Overall, the paper is well-written and brings together several recent works. The only comment I have is that there is a related line of work missing in the discussion. As mentioned above, Alabdulmohsin (2015) was an early work that proposed measuring the mutual information with a subset of size m=1 and showed that it was tight (in a certain sense). Based on that, a bound similar to Theorem 2.1 was derived for bounded losses and an equivalence relation was established with the VC dimension, which is similar to the works discussed in Section 4.2, in addition to concentration bounds. These are summarized in [2]. They should be included in the related works for completeness.

[1] Alabdulmohsin, "Algorithmic stability and uniform generalization." NeurIPS (2015).

[2] Alabdulmohsin, "Towards a Unified Theory of Learning and Information." Entropy 22.4 (2020): 438.

========

Post-rebuttal:

I have read the authors' response and I intend to keep my score as is. I find this paper a worthwhile contribution overall.

**Time Spent Reviewing:**

3

---

> ### Author Response · Authors · 2021-08-06
> **Response to Reviewer RQQn**
>
> > First, the authors present a theorem that generalizes a bound of Xu & Raginsky (2017) (Theorem 2.2 in the paper) in which the mutual information is measured between the weights W and the full dataset S. Theorem 2.2 uses a subset of S. However, I don't see how this theorem is different from Theorem 2.1 since taking a proper subset of a training set (i.e. for any fixed choice of u in the theorem), the distribution of S_u is the same as the distribution of a training sample of size m drawn iid. Would you please clarify why Eq 5 is not equivalent to Eq 4?
>
> Indeed, the distribution of $S_u$ is identical to the distribution of $m$ i.i.d. examples. Using only $m$ examples in Theorem 2.1 will give a bound of form $| \mathbb{E}\left[\mathcal{L}(A,S_u,R)-\mathcal{L}_\text{emp}(A,S_u,R)\right] | \le \sqrt{\frac{2\sigma^2 I(W; S_u)}{m}}$. Note that this bounds the generalization gap in the case of training with $m$ examples. Removing the absolute value, then taking expectation over $u$ and using the linearity of expectation will prove Eq. (5), which is nothing more than a slight generalization of the result of Xu and Raginsky [34]. In addition to this, Theorem 2.2 also bounds the expected squared generalization gap.
>
> > The authors then discuss the choice of the subset size m. They argue that having a smaller m yields a tighter bound. However, there is an early work that makes the same argument and is not discussed in the paper. In Alabdulmohsin (2015), m=1 is used to derive non-vacuous bounds and it is shown that those bounds were tight (see for instance [1] and the subsequent works summarized in [2]). Note that the difference between using the weights and using the function itself (decision boundary) was discussed there as well using data-processing. Since this is an earlier work that makes the same argument, it should be discussed.
>
> We thank the reviewer for the helpful references, we will discuss them in future revisions. The works are complementary to ours as they prove similar results (optimality of $m=1$ and effects of data processing) but for a different information measure: the mutual stability of [1].
>
> > The authors then discuss some implications. For example, they look into the case of ensemble learning and show that the f-CMI of an ensemble can be bounded using the f-CMI of the base models. However, this argument is not really useful because it does not explain why an ensemble method tends to generalize better. The bound for the ensemble is larger than the bound of any base classifier.
>
> It would be nice have a result that agrees with the practice where ensembling methods often generalize better. However, for this we would probably need to consider a concrete ensembling algorithm. The result of Sec 4.1 is for all possible ensembling algorithms. Among those there are ensembling algorithms which generalize worse than any of the individual algorithms. (Copying from the response to Reviewer 3S6u). The purpose of the result on ensembling algorithms is to demonstrate the compositionality the proposed f-CMI bound. One important property of this result is that it holds for any ensembling algorithm $g$ of the described form, no matter how complicated it is. Even when all individual algorithms generalize well, for a significantly complicated ensembling algorithm $g$ it might not be trivial to show that it also generalizes well (especially when $g$ is learned too, creating an extra overfitting source).
>
> >Finally, the authors validate the analysis with a few experiments. The authors argue that the f-CMI bound "closely tracks" the generalization gap but this is misleading because the difference is one order of magnitude and the fact it is decreasing with increasing sample size does not make it "tracking" the generalization gap. After all, all useful generalization bounds decrease with increasing sample size. I suggest that the authors rephrase that claim (perhaps by only arguing that the bound is non-vacuous). Also, in Line 319, the statement is false because weight-based mutual information bounds need not be vacuous (e.g. when m=1 as discussed earlier). Yes, they can be vacuous but need not be.
>
> We agree with the reviewer that "tracking closely" is an imprecise formulation. We used this wording relatively, having in our minds the fact that many generalization bounds for neural networks give very large upper bounds of generalization gap in practical settings of deep learning (large networks and not too large $n$). We will rephrase that and fix the false statement in Line 319.

---

### Official Review · Reviewer_3S6u · 2021-07-27

**Rating:** 7
**Confidence:** 4

**Summary:**

The paper continues a line of work which develops information-theoretic bounds on the generalization gap (i.e. the difference between e,pirical and population loss) in supervised learning.
Its main contributions are:
1. the authors introduce a natural “transductive” variant of the conditional mutual information bound (Section 3) and show that it is stronger than all previous bound. (Eg one can use it to prove that any proper learner for a VC class generalizes).
2. The authors quantitatively improve some of the previous bounds on input-output mutual information (theorem 2.2, 2.3), and on conditional mutual information (Theorem 2.6), by moving the expectation outside the square-root in the RHS of the inequality.


**Ethical Concerns:**

As far as I can see, there are none.

**Limitations And Societal Impact:**

I don't see any potential negative social impact.

**Main Review:**

Overall I liked this paper: I think it the transductive variant of the recently-well-studied conditional mutual information is natural, simple, and they show that this variant is in fact stronger.
Moreover, they also derive quantitative improvements over the previous bounds.

Some specific comments and questions to the authors:
1. I was not very convinced by the motivation using deep learning. The main reason is because the bounds studied in this paper imply that the empirical and population losses are close, and perhaps the greatest mystery DNN demonstrate is that they generalize, however they do not necessarily satisfy this property.
2. I would appreciate if the authors can elaborate more on their quantitative improvement over previous bound (by moving the expectation outside the square-root). Are there concrete, natural examples where it gives better bounds? How complicated is this improvement comparing to the previous (weaker) derivations?
3. In Theorem 4.1, please add that the learning algorithm is proper. Otherwise the statements seems false.
4. Can you please elaborate on the application wrt to ensemble methods? In which ways is it better than previous bounds?


**Time Spent Reviewing:**

3

---

> ### Author Response · Authors · 2021-08-06
> **Response to Reviewer 3S6u**
>
> > I was not very convinced by the motivation using deep learning. The main reason is because the bounds studied in this paper imply that the empirical and population losses are close, and perhaps the greatest mystery DNN demonstrate is that they generalize, however they do not necessarily satisfy this property.
>
> The bounds proposed in this paper are general and can be applied to approaches other than deep learning. In our applications we focused on deep learning as it is a challenging case. Indeed, in deep learning we often observe that training error is very close to zero, while test error is larger than zero. This creates a significant generalization gap. The proposed and existing information-theoretic bounds will result in proper upper bounds, and will not claim that training and test errors are close to each other. This can be used seen by interpreting these bounds differently. Instead of the standard interpretation, where we read these results as statements like "if the information terms are small, then training and test errors are close", we can read them as follows "if there is a large generalization gap, then information terms cannot be small, meaning that the network has to memorize some information about training data". Nevertheless, as we increase the number of examples $n$, the proposed generalization bound will converge to zero (this can be proven by showing that SGD becomes more stable as we increase $n$ and using it in Thm. 4.1), indicating that training and test errors will approach to each other. This happens either because the prediction problem gets fully solved or because the memorization capacity of the neural network gets exhausted.
>
>
> > I would appreciate if the authors can elaborate more on their quantitative improvement over previous bound (by moving the expectation outside the square-root). Are there concrete, natural examples where it gives better bounds? How complicated is this improvement comparing to the previous (weaker) derivations?
>
> By Jensen's inequality $\frac{1}{n}\sum_{i=1}^n\sqrt{a_i} \le \sqrt{\frac{1}{n}\sum_{i=1}^n a_i}$, where the equality holds when all $a_i$ are equal. To judge how large is the improvement, we need to know the distribution of $a_i$s, which in our case would be quantities like $I(W; Z_i)$ or $I(W; Z_i \mid Z_{-i})$. Estimating these quantities is challenging as both $W$ and $Z_i$ are usually high-dimensional non-discrete variables. Furthermore, the mutual information can be infinite. In Ref. [12] of the paper, the unique information $I(W; Z_i \mid Z_{-i})$ is upper bounded and estimated for some stochastic training algorithms. One relevant finding is that most examples have low information, while there is a long-tail of high-information examples (Fig. 1 of [12]). The distribution of these quantities is far from uniform and resembles a power law distribution. Therefore, one may expect a reasonable improvement when moving expectation outside square root. For example, let us assume that the distribution of $a_i$ follows the Zipf's law, where the number of elements $a_i=k, (1 \le k \le n)$ is proportional to $\frac{1}{k}$. The normalization constant will be $\Theta(\frac{\log n}{n})$. Then, $\frac{1}{n}\sum_{i=1}^n\sqrt{a_i} \approx \frac{1}{n} \sum_k \frac{n}{k \log n} \sqrt{k} = \Theta(\frac{\sqrt{n}}{\log n})$, while $\sqrt{\frac{1}{n}\sum_{i=1}^n a_i} = \sqrt{\frac{1}{n}\sum_k \frac{n}{k \log n} k} = \Theta(\frac{\sqrt{n}}{\sqrt{\log n}})$.
>
> We did not elaborate on the quantitative improvement arising when moving expectations outside the square root in section 2, as those results were not our main contributions. We will add more discussion on that in future revisions.
>
>
> > Can you please elaborate on the application wrt to ensemble methods? In which ways is it better than previous bounds?
>
> The purpose of the result on ensembling algorithms is to demonstrate the compositionality the proposed f-CMI bound. One important property of this result is that it holds for any ensembling algorithm $g$ of the described form, no matter how complicated it is. Even when all individual algorithms generalize well, for a significantly complicated ensembling algorithm $g$ it might not be trivial to show that it also generalizes well (especially when $g$ is learned too, creating an extra overfitting source).

---

### Official Review · Reviewer_LtsL · 2021-08-01

**Rating:** 5
**Confidence:** 3

**Summary:**

This paper studies information-based generalization bounds and derive several improved versions of existing generalization bounds using mutual information. The paper suggests a function-based conditional mutual information in order to tighten the existing generalization bounds and discusses some numerical results to support the shown generalization bounds.

**Limitations And Societal Impact:**

Please refer to my comments in the main review.

**Main Review:**

This work uses an information-theoretic approach to analyze the generalization properties of neural network classifiers. The work uses the mutual information between observed data and model parameters to bound the generalization error and then proves several variants of the existing information-based bounds through the application of conditional mutual information and function-based mutual information measures that focus on the classifier's output instead of the input samples.

In general, the paper tries to address an important problem in learning theory on the generalization properties of overparameterized deep neural networks. The paper's analysis seems to improve the existing mutual information-based generalization bounds. However, I am not completely sure how to estimate the bounds from real data because the paper has no guarantees that the estimation error for the information measures is bounded. In addition, the paper contains many theoretical statements, which makes it hard for a reader to focus on and understand the paper's main contribution.

To further explain the comments above, my main concern with this work is how to evaluate the generalization bounds for standard datasets and deep neural network classifiers. The generalization bounds include conditional mutual information terms which are hard to evaluate from real data in practical settings. Also, the upper bound in Corollary 2 includes n mutual information terms where for every term we have only one observed sample. Therefore, it is not clear how one should estimate every mutual information term from only one sample.

In addition to the estimation of information measures, the current paper includes too many theoretical statements that makes it hard to distinguish the paper's main contribution from the existing results in the literature. I highly suggest removing the less important results from the main text and further explaining how the generalization bounds in this work can be more useful than existing information-based generalization bounds. The current draft seems to only suggest that the new bounds rely on conditional information measures, while this new feature may not be that useful in practice because the conditional information measures are statistically unaffordable to estimate.

**Time Spent Reviewing:**

5

---

> ### Author Response · Authors · 2021-08-06
> **Response to Reviewer LtsL**
>
> >I am not completely sure how to estimate the bounds from real data because the paper has no guarantees that the estimation error for the information measures is bounded.
> > To further explain the comments above, my main concern with this work is how to evaluate the generalization bounds for standard datasets and deep neural network classifiers. The generalization bounds include conditional mutual information terms which are hard to evaluate from real data in practical settings. Also, the upper bound in Corollary 2 includes n mutual information terms where for every term we have only one observed sample. Therefore, it is not clear how one should estimate every mutual information term from only one sample.
>
> The main result of this paper is the bound described in Corollary 2 (eq. 21). Computing this bound involves estimating conditional mutual informations between 2 predictions and binary variable, conditioned on the choice of training and test examples $\tilde{Z}$. For each particular value of $\tilde{Z}=z$ this estimation is often easy to do. For example, in case of binary classification one needs to estimate mutual information terms of form $I([F_1, F_2]; S_i)$, where $F_1, F_2$ and $S_i$ are all binary variables. This can be done straightforwardly by estimating all the 8 states of the joint distribution  $p(f_1,f_2,s_i)$ and using them in a plug-in estimator. The estimation bias in this case is $O(\frac{1}{n})$, while the variance is $O(\frac{(log n)^2}{n})$, where $n$ is the number of samples used in the plug-in estimator [1]. In the next step of estimation, where we approximate expectation over $\tilde{Z}$ using $n'$ samples, the bias stays $O(\frac{1}{n})$, while the variance gets increased by $O(\frac{1}{n'})$.
>
> Note that estimating other information-theoretic bounds is significantly harder. The bounds of Xu and Raginsky [ref. 34 in the paper], Negrea et el. [ref. 19 in the paper], and Bu et al. [ref. 4 in the paper] are hard to estimate as they involve estimation of mutual information between a high-dimensional non-discrete variable $W$ and at least one example $Z_i$. Furthermore, this mutual information can be infinite in case of deterministic algorithms or when $H(Z_i)$ is infinite.
> The bounds of Haghifam et al. [ref. 11 in the paper] and  Steinke and Zakynthinou [ref. 28 in the paper] are also hard to estimate as they involve estimation of mutual information between $W$ and at least one train-test split variable $S_i$.
>
> We will add more discussion on estimation of our main bound and challenges arising in estimation of existing information-theoretic bounds.
>
> >In addition, the paper contains many theoretical statements, which makes it hard for a reader to focus on and understand the paper's main contribution.
> > In addition to the estimation of information measures, the current paper includes too many theoretical statements that makes it hard to distinguish the paper's main contribution from the existing results in the literature. I highly suggest removing the less important results from the main text and further explaining how the generalization bounds in this work can be more useful than existing information-based generalization bounds. The current draft seems to only suggest that the new bounds rely on conditional information measures, while this new feature may not be that useful in practice because the conditional information measures are statistically unaffordable to estimate.
>
> We agree with the reviewer that the current manuscript is dense and can be hard to read. This arises from the particular structure we chose. In section 2 we describe existing weigh-based information-theoretic generalization bounds, slightly improve some of them, and prove some relations between them. We do not consider the results of this section as our main contribution. The purpose of this section is to introduce existing bounds, show that they can be derived in unified fashion, and prepare grounds for the f-CMI bounds introduced in section 3, which we consider our main contribution. The statements in section 4 are various applications of the f-CMI bounds, aimed towards understanding how the proposed bounds can be more useful than existing bounds. In particular, in Sec. 4.2 we show that the proposed f-CMI bounds give better results in case of binary classification with finite VC dimension hypothesis classes; and in Sec. 4.3 we show that the proposed bounds allow extensions to stable deterministic or stochastic algorithms.
>
> We will clarify our contributions and explain the structure of the paper better in future revisions.
>
>
>
> ### References
> [1] Paninski, Liam. "Estimation of entropy and mutual information." Neural computation 15.6 (2003): 1191-1253.

---

### Decision · Program_Chairs · 2021-09-27

**Decision:**

Accept (Poster)

**Comment:**

The paper continues a line of work which develops information-theoretic bounds on the generalization gap (i.e. the difference between e,pirical and population loss) in supervised learning. Almost all reviewers were favorable for acceptance, and found the results innovative.


Please address reviewers comments in the final version.